# UNSEEN IMAGE SYNTHESIS WITH DIFFUSION MODELS

## ABSTRACT

While the current trend in the generative field is scaling up towards larger models and more training data for generalized domain representations, we go the opposite direction in this work by synthesizing *unseen* domain images *without additional training*. We do so via latent sampling and geometric optimization using pre-trained and frozen Denoising Diffusion Probabilistic Models (DDPMs) on single-domain datasets. Our key observation is that DDPMs pre-trained even just on single-domain images are already equipped with sufficient representation abilities to reconstruct arbitrary images from the inverted latent encoding following bi-directional deterministic diffusion and denoising trajectories. This motivates us to investigate the statistical and geometric behaviors of the Out-Of-Distribution (OOD) samples from unseen image domains in the latent spaces along the denoising chain. Notably, we theoretically and empirically show that the inverted OOD samples also establish Gaussians that are distinguishable from the original In-Domain (ID) samples in the intermediate latent spaces, which allows us to sample from them directly. Geometrical domain-specific and model-dependent information of the unseen subspace (*e.g.*, sample-wise distance and angles) is used to further optimize the sampled OOD latent encodings from the estimated Gaussian prior. We conduct extensive analysis and experiments using pre-trained diffusion models (DDPM, iDDPM) on different datasets (AFHQ, CelebA-HQ, LSUN-Church, and LSUN-Bedroom), proving the effectiveness of this novel perspective to explore and re-think the diffusion models' data synthesis generalization ability. [1]

## 1 INTRODUCTION

*"Seek inward in face of difficulties." – Mencius*

Generalization has always been a challenge in data synthesis. The current research trend focuses on leveraging larger models and more training data as to facilitate improved generalization. The popularity of recent large-scale models such as DALLE-2 (Ramesh et al., 2022) and Imagen (Ho et al., 2022a) have demonstrated the impressive and promising representation ability when trained on an enormous amount of data. However, the approach of scaling up may ultimately hit its limit as the available data is still finite; further, scaling up requires enormous resources, severely limiting the number of research groups that are able to participate and contribute to the work, and consequently hindering research progress. Thus, we look inwards to investigate an alternative direction in this work. Specifically, we propose to leverage the powerful yet under-explored potential of the high-dimensional latent spaces in pre-trained denoising diffusion probabilistic models (DDPMs) (Ho et al., 2020; Nichol & Dhariwal, 2021) on single-domain datasets, to generate images in unseen domains *without learning any extra neural networks* to explicitly model the unseen data distributions.

This work is built upon our **key observation**: as the state-of-the-art generative models, a pre-trained DDPM on single-domain images (*e.g.*, dog faces) already has sufficient representation ability to reconstruct arbitrary unseen images (*e.g.*, human faces and churches) from the OOD latent encodings, as shown in Fig. 1. However, we also argue that the reconstruction ability of unseen domain images is subject to the prerequisite of the deterministic inversion (diffusion) and denoising trajectories. Specifically, we employ the deterministic image inversion technique derived from the denoising diffusion implicit models (DDIMs) (Song et al., 2021a) to convert those raw images from unseen

---

[1]This work includes abundant supporting analysis, qualitative examples and discussions in the supplementary. Core code will be submitted to AC during the discussion, complete code will be released upon acceptance.

Figure 1: **Unseen domain image synthesis with a pre-trained diffusion generative model on AFHQ-Dog-256 (Choi et al., 2020) dataset, without any additional training to adapt to the new domains.** In the second column, we observe that a DDPM pre-trained on dog faces nevertheless has sufficient representation ability to accurately reconstruct arbitrary images. Leveraging this unique ability and our comprehensive analysis of the unseen OOD distributions in the latent spaces, we can even *generate* images of different domains (*e.g.*, human faces, churches, bedrooms, and cats) from this model leveraging our proposed *UnseenDiffusion* method, as shown in the third column.

domains into the latent encodings, and use the pre-trained DDPMs to reconstruct the images from the inverted latent encodings via a *relatively deterministic* denoising process [2], as detailed in Sec. 3.2.

The above observation motivates re-thinking the representation and generalization ability of diffusion models, and to leverage this unique property to synthesize unseen images via pre-trained and frozen DDPMs in this work. Recall the conventional pipeline for image synthesis via generative models (*e.g.*, VAEs (Kingma & Welling, 2014), GANs (Goodfellow et al., 2014), and DMs (Ho et al., 2020)), which usually consists of distribution learning, latent sampling and decoding. While the most natural way seems to be learning the unseen latent distributions via extra neural networks as most existing works do (Zhou et al., 2020; 2021; Wang et al., 2019), we tackle the problem without using networks to learn the unknown distributions. Sampling from an unknown distribution in high-dimensional space without prior knowledge is extremely difficult. However, by revisiting the theoretical non-Markovian formulation of DDIMs (Song et al., 2021a), we show that the inverted OOD latent encodings also follow a Gaussian distribution *in theory*, like the original ID latent variables along the denoising chain. Based on the theoretical support, we estimate the implicit latent Gaussian distributions of the OOD images for synthesizing new samples, as specified in Sec. 3 and Appendix C.

On the other hand, we acknowledge that a Gaussian prior alone is insufficient for generating new unseen images in practice, due to the following two main reasons: *First,* there always exists a gap between theory and practical model training (*i.e.*, a direct sampling from the estimated OOD Gaussian may not be precise enough to characterize the actual probability concentration mass in latent spaces given different base models). *Second,* the sampled latent encoding may be "captured" by the ID denoising trajectories given the bandwidth constraint, which leads to a typical failure case of generating *in-distribution* images instead of the target unseen images (see Fig. 4). To tackle the above empirical challenges, we gain inspiration from recent works (Zhu et al., 2023a) to rethink the generation process from a geometrical and spatial point of view, and propose to further optimize the

---

[2]We use the term "relatively deterministic" to represent the fact that the denoising trajectory for unseen image synthesis should have a non-zero bandwidth to tolerate some degree of stochasticity, which is a critical property in practice explained in Sec. 3.2.

sampled latent OOD encodings by rejecting those that do not satisfy the geometric constraints (*e.g.*, pair-wise distance and angles). Interestingly, we observe that the inverted OOD latent encodings exhibit consistent geometric characteristics in the intermediate high-dimensional latent spaces, which in return support our theoretical Gaussian assumptions. The proposed geometrical optimization can be considered as supplementary model-dependent and domain-specific knowledge that helps to achieve the challenging synthesis task, with details explained in Sec. 3 and Appendix D.

We conduct experiments using various DDMs (DDPM (Ho et al., 2020), improved DDPM (Nichol & Dhariwal, 2021)) and datasets (CelebA-HQ (Karras et al., 2017), AFHQ-Dog (Choi et al., 2020), LSUN-Church (Yu et al., 2015), LSUN-Bedroom (Yu et al., 2015)), achieving promising performance for this challenging task of unseen domain image synthesis, as demonstrated in Fig. 1. Notably, our work also reveals an unusual fact that **contradicts** the findings of previous learning-based domain generalization works (Zhou et al., 2020; 2021; Wang et al., 2019), which believe that it is usually easier to generalize model abilities to unseen domains similar to the trained ones. However, we show that our proposed method achieves better performance in dramatically different unseen domains in Sec. 4. Overall, we name our entire framework *UnseenDiffusion*, and believe that our interesting empirical observations, theoretical findings, extensive experimental analysis and novel perspective to rethink the domain generalization ability in pre-trained DDMs in this work shall bring future insights that benefit our research community [3].

## 2 RELATED WORK

**Denoising Diffusion Models.** Denoising Diffusion Models (DDMs) (Sohl-Dickstein et al., 2015; Ho et al., 2020) are the state-of-the-art generative models for data synthesis in images (Ramesh et al., 2022; Rombach et al., 2022; Nichol & Dhariwal, 2021; Gu et al., 2022; Zhu et al., 2023b; Dhariwal & Nichol, 2021; Hu et al., 2021), videos (Ho et al., 2022b), and audio (Kong et al., 2020; Zhu et al., 2023b; Mittal et al., 2021). Inspired by thermodynamics, the core design of DDMs consists of a Markov chain in two directions: the forward diffusion process and the reverse denoising process. The diffusion process gradually adds stochastic Gaussian noises to the data sample $x_0$ at each step, converting the data into a noisy latent encoding at the end of the Markov chain at step $T$. In contrast, the denoising process seeks to remove the noises added and to restore the initial data. This gradual transition process models a random walk in the high-dimensional latent spaces and forms a trajectory along the denoising chain (Song & Ermon, 2020; Song et al., 2020).

As the denoising trajectory proceeds, the latent in-domain distribution goes from a pre-defined standard Gaussian to the learned data distribution. Part of existing works consider this process from the perspective of score-based functions (Kingma et al., 2021; Song et al., 2021b; Huang et al., 2021; Vahdat et al., 2021), but few have touched the topic of unseen distributions within learned intermediate latent spaces.

**Domain Generalization.** Domain Generalization (Wang et al., 2022a) that aims to generalize deep learning model to unseen distributions has been an important research topic in broad machine learning area (Ganin et al., 2016; Zhao et al., 2020; Zhou et al., 2021; Muandet et al., 2013; Li et al., 2017) with various computer vision applications such as recognition (Peng et al., 2019; Rebuffi et al., 2017), detection (Zhang et al., 2021) and segmentation (Hoffman et al., 2018; Gong et al., 2019). In the generative field, it even becomes a more challenging task, with the extra demand to sample from the generalized distributions. Classic approaches seek to extend the representation ability of generalized domains by training extra neural network modules (Zhou et al., 2020; 2021; Wang et al., 2019). More recent trend in computer vision community is scaling up the model and dataset sizes as the most intuitive and obvious solutions (Ramesh et al., 2022; Ho et al., 2022a; Rombach et al., 2022).

In this work, we tackle the challenge from a novel perspective by exploring the potential of latent space from a relatively small generative model trained on the single-domain dataset.

**Latent Space of Deep Generative Models.** Understanding the latent space of generative models (Karras et al., 2017; Abdal et al., 2019; Gal et al., 2022) help to benefit popular downstream tasks such as data editing and manipulation (Zhu et al., 2016; Shen et al., 2020; Kwon et al., 2023; Zhu et al., 2023a; 2020). A large portion of works has been exploring this problem within the context of GAN inversion (Xia et al., 2022), where the typical methods can be mainly divided into

---

[3] Please see Appendix A for some discussions on the high-level insights for several open questions.

either learning-based (Zhu et al., 2016; Richardson et al., 2021; Wei et al., 2022; Alaluf et al., 2021) or optimization-based categories (Abdal et al., 2019; 2020; Huh et al., 2020; Creswell & Bharath, 2018). More recently, with the growing popularity of diffusion generative models, researchers have also focused on the latent space understanding of DDMs for better synthesis qualities or semantic control (Rombach et al., 2022; Kwon et al., 2023; Zhu et al., 2023a; Yang et al., 2023).

Our work is also related to the latent space understanding of DDMs, and we leverage the geometric properties and apply them to the difficult unseen domain image synthesis task.

## 3 UNSEEN IMAGE SYNTHESIS

In this section, we describe our methodology design to generate images of unseen domains using a pre-trained and frozen diffusion model via latent OOD sampling and geometric optimizations, without training any extra neural networks to explicitly model the unknown distributions.

### 3.1 PROBLEM FORMULATION

Given a pre-trained $T$-step DDM $p$ (we omit $\theta$ from the more commonly used notation $p_\theta$ since we use frozen parameters in this work) trained on a single-domain dataset with dimensionality $d$ equal to the total resolution of images, our objective is to synthesize unseen images $\mathbf{x}^{out}$ different from the training domain. We denote the learned ID distribution in the $\epsilon_t$ latent space as $\phi_t^{in}$, where $t \in \{T, ..., 0\}$ is the diffusion step. Similarly, we define a target unseen image domain with unknown latent distribution $\phi_t^{out}$ in $\epsilon_t$. Suppose that we have $N$ raw images from the target unseen domain, thus we can obtain $N$ latent OOD encodings $\mathbf{x}_{inv,t}^{out}$ at each latent space $\epsilon_t$ via the deterministic inversion method derived from DDIMs Song et al. (2021a). Our objective is to deduce the behavior of the unknown latent distribution $\phi_t^{out}$ based on $N$ OOD samples, either raw images $\mathbf{x}^{out}$ or inverted ones $\mathbf{x}_{inv,t}^{out}$, and eventually sample $\mathbf{x}_{sample,t}^{out}$ from $\phi_t^{out}$ to synthesize images $\mathbf{x}^{out}$ of the target unseen domain using the frozen model $p$. We further use $p_s$ and $p_i$ to represent the stochastic (Ho et al., 2020) and deterministic (Song et al., 2021a) generation processes, respectively. In addition, we use the hyper-parameter $\eta$ (Song et al., 2021a) to characterize the degree of stochasticity in the generative process, with $\eta = 1$ for $p_s$ and $\eta = 0$ for $p_i$. At intermediate stochastic levels, we adopt the notation $p_{\eta=k}$ with $k$ equals a constant between 0 and 1.

### 3.2 REPRESENTATION ABILITY FOR UNSEEN DOMAINS

**Unseen Domain Image Reconstruction.** Our fundamental key observation is that a DDPM trained even on a single-domain small dataset already has sufficient representation ability to accurately reconstruct arbitrary unseen images from the inverted latent encoding $\mathbf{x}_{inv,t}^{out}$ following a deterministic denoising trajectory $p_i$, as shown in Fig. 1. We further argue that this representation ability is subject to the deterministic inversion and denoising techniques we adopted from Song et al. (2021a). Note the inverted space can be an arbitrary step $t$ along the chain for reconstruction purposes.

Specifically, DDIMs (Song et al., 2021a) propose a different deterministic sampling process, which considers a family of inference distribution as:

$$\mathbf{x}_{t-1} = \sqrt{\alpha_{t-1}}\left(\frac{\mathbf{x}_t - \sqrt{1-\alpha_t}\epsilon_t^\theta(\mathbf{x}_t)}{\sqrt{\alpha_t}}\right) + \sqrt{1-\alpha_{t-1}-\sigma_t^2} \cdot \epsilon_t^\theta(\mathbf{x}_t) + \sigma_t z_t, \tag{1}$$

where $\sigma_t = \eta\sqrt{(1-\alpha_{t-1})/(1-\alpha_t)}\sqrt{1-\alpha_t/\alpha_{t-1}}$, as the $\eta$ controls the variance of Gaussian transition kernel and thus characterizes degree of stochasticity. A variant of the above process allows us to derive a deterministic inversion technique by connecting the above Eqn. 1 to the neural ordinary differential equations (ODEs), similar as in previous diffusion based data editing works (Kim et al., 2022; Kwon et al., 2023; Zhu et al., 2023a). More details about the deterministic inversion and denoising methods are included in the Appendix C.

**Bandwidth of Unseen Denoising Trajectory.** In practice, the representation ability in the latent space itself is not adequate to synthesize new images for unseen domains; there exists another implicit yet critical precondition to leverage the discovered feature of a pre-trained DDPM for unseen image synthesis, which we refer to as "bandwidth of the unseen trajectories," denoted as $\mathcal{B}_{\eta,t}$. In other words, in order to generate plausible unseen images from a sampled latent OOD

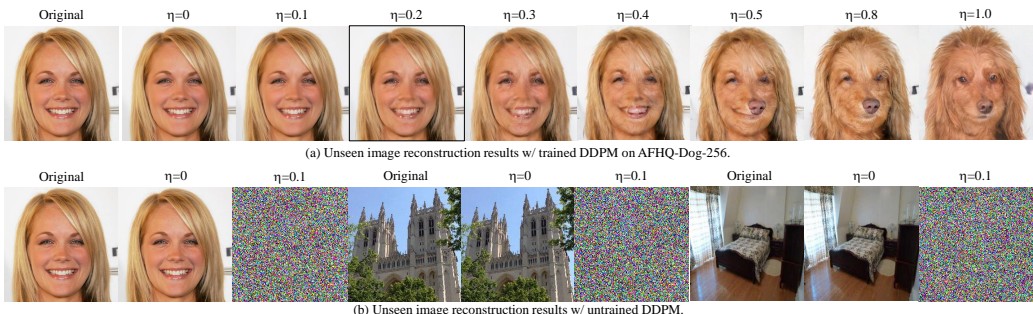

Figure 2: **Illustration of the unseen trajectory bandwidth at the latent space $\epsilon_t$.** With a trained DDPM, we can reconstruct unseen images from new domains with a certain stochasticity tolerance (top row). In contrast, the untrained DDPM can also do the reconstruction, but with extremely limited bandwidth for the unseen trajectories. We empirically choose bandwidth $\mathcal{B}_\eta$ for the target unseen domain using the maximum value of $\eta$ that guarantees the reconstruction quality (*e.g.*, $\mathcal{B}_\eta = 0.2$, $t = 500$ in this example).

encoding $\mathbf{x}_t^{out} \in \phi_t^{out}$, the actual denoising trajectory should tolerate a certain range of stochasticity, following a "relatively deterministic" denoising process $p_{\eta=k}$, with $k \neq 0$, instead of the completely deterministic $p_i$. Another interpretation is to analogue the trajectory bandwidth $\mathcal{B}_{\eta,t}$ to the actual subspace volume occupied by the OOD latent samples. In the extreme case where the model is untrained, we can still invert and reconstruct arbitrary images following a deterministic trajectory. However, this untrained model can not be used for actual generation purposes because those latent encodings establish no generalizable patterns but simply map an arbitrary bijective function between the latent and image spaces. Fig. 2 illustrates this concept of bandwidth for unseen trajectories; we note that the bandwidth is an important empirical parameter for our unseen image synthesis task that depends on the pre-trained models, the specific unseen target domains, as well as the diffusion steps. More discussions are included in Sec. 4.2 and Appendix F.3. In this work, we define the bandwidth $\mathcal{B}_{\eta,t}$ as the maximum empirical value of $\eta$ that guarantees the reconstruction quality for unseen images at the latent space $\epsilon_t$.

### 3.3 *UnseenDiffusion* FOR IMAGE SYNTHESIS

After having verified the representation ability via unseen image reconstructions, we seek to gain more knowledge on the unseen distribution established by the OOD latent encodings, as to be able to generate novel images from the unseen domain.

**Non-Markovian Inversion.** We propose to take a closer look at what the *forward diffusion process* actually does in theory to better understand the hidden properties of those latent encodings. The core idea of DDIMs (Song et al., 2021a) modifies the original forward diffusion process from a Markov process to a non-Markov one, by directly adding the information from $\mathbf{x}_0$ in inferring $\mathbf{x}_t$, which changes the initial diffusion from $q(\mathbf{x}_t|\mathbf{x}_{t-1})$ to $q(\mathbf{x}_t|\mathbf{x}_{t-1}, \mathbf{x}_0)$. The theoretical rationale behind is that the DDPM objective only depends on the marginals $q(\mathbf{x}_t|\mathbf{x}_0)$ instead of directly on the joint $q(\mathbf{x}_{1:T}|\mathbf{x}_0)$, with the intuitive motivation to accelerate the generation.

The above formulation implies that this revised non-Markovian diffusion process satisfies the below family $\mathcal{Q}$ of inference distributions indexed by a real vector $\sigma \in \mathbb{R}_{\geq 0}^T$:

$$q_\sigma(\mathbf{x}_t|\mathbf{x}_0) = \mathcal{N}(\sqrt{\alpha_t}\mathbf{x}_0, (1-\alpha_t)\mathbf{I}), \qquad (2)$$

where $\alpha_t$ [4] comes from a pre-defined sequence. In addition, the Eqn. 2 simultaneously ensures that $q(\mathbf{x}_t|\mathbf{x}_{t-1}, \mathbf{x}_0)$ is also Gaussian for all $t > 1$ as below:

$$q_\sigma(\mathbf{x}_{t-1}|\mathbf{x}_t, \mathbf{x}_0) = \mathcal{N}(\sqrt{\alpha_{t-1}}\mathbf{x}_0 + \sqrt{1-\alpha_{t-1}-\sigma_t^2} \cdot \frac{\mathbf{x}_t - \sqrt{\alpha_t}\mathbf{x}_0}{\sqrt{1-\alpha_t}}, \sigma_t^2 I). \qquad (3)$$

The detailed proofs and derivations of Eqn. 2 and Eqn. 3 can be found in Appendix C. Particularly, we note that the above derivations in the diffusion forward direction do not touch the actual denoising

---

[4]We adopt the definition of $\alpha_t$ from DDIMs, which is slightly different from DDPMs (Ho et al., 2020) as detailed in Appendix C.

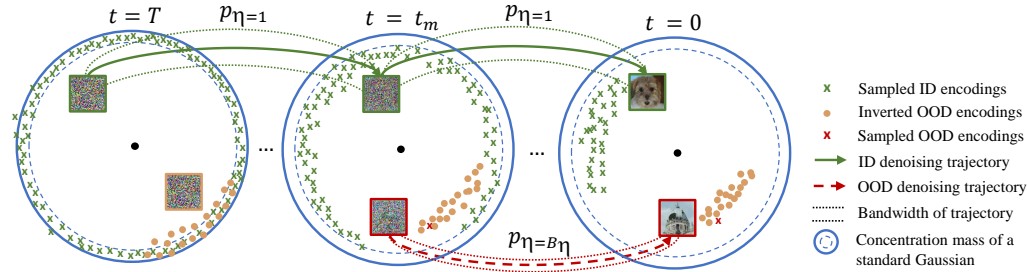

Figure 3: **Illustration of geometric distributions for ID (*e.g.*, dogs) and OOD (*e.g.*, church) latent encodings along the denoising chain.** At the departure latent space at step $T$, the ID distribution is by definition a standard Gaussian in high-dimensional space, whose concentration mass is a thin annulus (Blum et al., 2020). As the denoising chain proceeds, the ID and OOD Gaussian become separable at the mixing step $t_m$ (Zhu et al., 2023a), which allows us to perform the latent geometric sampling for unseen OOD domains. Following a relatively deterministic denoising process $p_{\eta=\mathcal{B}_\eta}$ with the trajectory bandwidth $\mathcal{B}_\eta$, we can finally synthesize images for the target unseen domain. Note that the ID trajectory always has a larger bandwidth than the OOD trajectory.

process $p$. In other words, the distribution in the intermediate latent spaces always satisfies Eqn. 2 *in theory*, and is not model dependent, conditioned on the actual raw data $x_0$.

While we have theoretically shown that the family $\mathcal{Q}$ of inference distributions matches the same marginal as described in Eqn. 2, the mean values of the Gaussians are data-dependent, and can only be empirically decided. Therefore, we use the $\mathbf{x}^{out}$ for estimating the mean value for unseen images [5].

**Latent Sampling with Geometric Optimization.** Given an estimated Gaussian prior for the unseen images is not sufficient to synthesize new *unseen* images. We propose that there exist two key factors for a successful trial to generate an unseen image: *"escape the ID trajectory of trained models"* and *"stay close to the OOD trajectory of target unseen images"*. As demonstrated in Fig. 4, there exists **two typical failure cases** for unseen image synthesis with a given DDPM: the generated image $\mathbf{x}_0^{out}$ become *in-distribution*, or the generated image has very low visual quality. The cause for the first failure case is that the OOD denoising trajectories get interfered by the ID ones. In comparison, the low-fidelity case is due to the fact the denoising trajectory gets too "off-road" despite not being entirely captured by the ID ones.

Inspired by Zhu et al. (2023a) where the authors tackle the semantic editing problem from a geometric point of view by studying the high-dimensional latent spaces, we note both above causes can be mitigated by additional geometric restrictions. Interestingly, we consistently observe the following geometric properties that can be used as rejection criteria for optimizing the latent sampling, after testing various sets of ID and OOD combinations with different pre-trained diffusion models. They also serve as additional empirical validation for the Gaussian assumption of inverted OOD samples.

**Observation 3.1** For any OOD sample pairs $\mathbf{x}_{inv,i}^{out}$ and $x_{inv,j}^{out}$ from the sample set, the Euclidean distance between these two points is approximately a constant $d_o$.

**Observation 3.2** For any three OOD samples $\mathbf{x}_{inv,i}^{out}$, $\mathbf{x}_{inv,j}^{out}$ and $\mathbf{x}_{inv,k}^{out}$ from the sample set, the angle formed between $\vec{\mathbf{x}_{inv,k}^{out}\mathbf{x}_{inv,i}^{out}}$ and $\vec{\mathbf{x}_{inv,k}^{out}\mathbf{x}_{inv,j}^{out}}$ is always around $60°$.

**Observation 3.3** For any OOD sample pairs $\mathbf{x}_{inv,i}^{out}$ and $\mathbf{x}_{inv,j}^{out}$, and the high-dimensional origin $O$, the angle formed between $\vec{O\mathbf{x}_{inv,i}^{out}}$ and $\vec{O\mathbf{x}_{inv,j}^{out}}$ is always around $90°$.

All these three observations above are typical geometric properties possessed by an isotropic high-dimensional Gaussian. We acknowledge that it is usually very challenging to deduce an unknown high-dimensional distribution solely based on its geometric properties and that there may exist other complex distributions that exhibit similar behaviors we have observed. However, it appears from our theoretical and empirical analysis, that the OOD Gaussian assumption holds.

---

[5]There are alternative ways for estimating the Gaussian, which are more empirical-driven, see details in Appendix E.2.

Table 1: **Geometric properties of different OOD domains from the latent space at the mixing step (Zhu et al., 2023a).** The results are computed based on 1K sample pairs, we report the mean and std for each geometric measurement.

| Domain | Pair-Distance | Pair-Angle | Angle to Origin |
|---|---|---|---|
| Dog(ID) | 608.3±3.2 | 60.0±0.4 | 90.1±0.3 |
| CelebA(OOD) | 577.1±5.0 | 60.0±0.5 | 89.7±0.4 |
| Cat(OOD) | 570.8±4.5 | 60.0±0.4 | 89.8±0.3 |
| Church(OOD) | 568.2±6.5 | 60.0±0.7 | 89.8±0.4 |
| Bedroom(OOD) | 574.3±5.5 | 60.0±0.6 | 89.5±0.5 |

Table 2: **Reconstruction results and the empirical bandwidth for different domains.** We use an iDDPM trained on AFHQ-Dog and 1K testing images to compute the reported scores.

| Method | Recons. Domain | MAE ($\downarrow$) | $\mathcal{B}_\eta$ ($\uparrow$) |
|---|---|---|---|
| pSp Richardson et al. (2021) | CelebA (ID) | 0.079 | - |
| e4e Tov et al. (2021) | CelebA (ID) | 0.092 | - |
| ReStyle Alaluf et al. (2021) | CelebA (ID) | 0.089 | - |
| HFGI Wang et al. (2022b) | CelebA (ID) | 0.062 | - |
| Ours *UnseenDiffusion* | Dog (ID) | 0.073 | 1 |
| | CelebA (OOD) | 0.073 | 0.2 |
| | Church (OOD) | 0.074 | **0.3** |
| | Bedroom (OOD) | 0.072 | **0.3** |

To sum up, the *UnseenDiffusion* pipeline (see Algo. 3) mainly consists of estimating the unseen Gaussian using Eqn. 2, computing the geometric properties via Algo. 1, conducting the geometric optimization via Algo. 2, and denoising the latent using frozen DDPMs $p$ with bandwidth $\mathcal{B}_\eta$. The concrete algorithms are included in Appendix F due to space limitations.

# 4 EXPERIMENTS

## 4.1 EXPERIMENTAL SETUP

**Model Zoos and Datasets.** We adopt pre-trained DDPMs on different single domain datasets as our base models for experiments: improved DDPM (Nichol & Dhariwal, 2021) trained on AFHQ-Dog dataset (Choi et al., 2020), and DDPM (Ho et al., 2020) trained on the CelebA-HQ dataset (Karras et al., 2017), the LSUN-Church dataset (Yu et al., 2015), and the LSUN-Bedroom dataset (Yu et al., 2015). Each model generates images in the resolution of $256^2$, resulting in the total dimensionality for the latent spaces to be $d = 196, 608$.

**Evaluations and Comparisons.** For the reconstruction task, we calculate the Mean Absolute Error (MAE) as quantitative metrics. We compare the results with other popular GAN based methods, such as pSp (Richardson et al., 2021), e4e (Tov et al., 2021), ReStyle (Alaluf et al., 2021) and HDGI (Wang et al., 2022b). The GAN-based models usually do not have the ability of unseen image reconstruction [6].

For the unseen image synthesis task, we mainly use the FID score (Heusel et al., 2017) as the quantitative metric as in most generative works. We report the in-domain FID scores obtained via the generative models as the upper bound on each domain (*i.e.*, each dataset). In the meanwhile, we also compare several different SOTA image-to-image translation methods via diffusion models as baselines, as they also output images in domains that are different to the trained ones. However, it is worth noting that those baseline methods (*i.e.*, EGSDE (Zhao et al., 2022), DiffusionClip (Kim et al., 2022), and Asyrp (Kwon et al., 2023)) are not really generating unseen images, but rather editing original ID image to a target unseen domain. Moreover, those are learning-based methods trained on each unseen domain, while our method operates on a single mutual latent space from the pre-trained base diffusion models.

**Implementation Details.** For all the experiments, we use the technique of skipping diffusion steps to accelerate the inversion and denoising process as in previous works (Song et al., 2021a; Kwon et al., 2023; Zhu et al., 2023a) without evident perceptual impairment to the image quality. Specifically, both inversion and denoising trajectories include 60 steps. The inversion process goes through 60 steps in total from $t = 0$ to $t_m = 500$ with a uniform skip interval; the denoising process follows the same steps in the reverse direction. We use in general 2K OOD samples to estimate the unseen Gaussian distributions. The number of randomly picked reference OOD samples for geometric optimization $\mathcal{N}_{ref}$ is chosen to be 4, meaning that we compare the sampled latent encoding with four inverted encodings in terms of their geometric properties before rejecting or accepting this sample.

---

[6]This is important to distinguish the generation problem of DDPMs and GANs, with detailed discussion within the topic of *"source of generalization ability"* in Sec. 4 and Appendix A

Table 3: **Image synthesis results for different domains.** We include three groups of comparisons from top to bottom. *L* stands for learning-based methods. The first group shows the results for ID synthesis via different generative models as *upper bounds*. The second group lists several learning-based methods using the DDM as the base model but perform the image-to-image (I2I) translation task. The third group include the results for our method using different DDMs as base models.

| Method | Category | Dog | CelebA | Church | Bedroom |
|---|---|---|---|---|---|
| StackGAN++ (Zhang et al., 2018) | L&ID | - | - | - | 35.61 |
| VQ-GAN (Esser et al., 2021) | L&ID | - | 10.2 | - | - |
| StyleGAN2 (Karras et al., 2020) | L&ID | - | - | - | 11.52 |
| DDPM (Ho et al., 2020) | L&ID | 7.74 | 8.58 | 11.20 | 10.89 |
| DDIM (Song et al., 2021a) | L&ID | 7.26 | 7.89 | 10.88 | 6.80 |
| iDDPM (Nichol & Dhariwal, 2021) | L&ID | 6.33 | 6.94 | 8.86 | 6.79 |
| EGSDE (Zhao et al., 2022) | L&I2I | 51.04 | - | - | - |
| DiffusionClip (Kim et al., 2022) | L&I2I | *ID* | 43.6 | 66.3 | 68.1 |
| Asyrp (Kwon et al., 2023) | L& I2I | *ID* | **38.7** | 59.5 | 57.1 |
| Ours-DDPM | OOD | 48.8 | *ID* | 46.7 | 45.9 |
| Ours-DDPM | OOD | 47.6 | 47.2 | *ID* | 46.8 |
| Ours-DDPM | OOD | **47.3** | 46.9 | 47.1 | *ID* |
| Ours-iDDPM | OOD | *ID* | 43.5 | **43.9** | **43.4** |

## 4.2 EXPERIMENTAL RESULTS

**Reconstruction Results.** We show the evaluation results for the reconstruction test in Tab. 2, where we demonstrate that a diffusion model pre-trained on a single-domain image dataset already has sufficient representation ability to reconstruct new domain images following the deterministic inversion and relatively deterministic denoising trajectories.

Specifically, we also report the bandwidth $\mathcal{B}_\eta$ we have empirically obtained for different unseen domains using a pre-trained iDDPM (Nichol & Dhariwal, 2021) on the AFHQ-Dog dataset (Choi et al., 2020). While this bandwidth is an empirical parameter depending on the pre-trained base DDM, it reveals **an interesting fact that contradicts** conventional wisdom. While classic learning-based methods for domain generalization (Zhou et al., 2020; 2021; Wang et al., 2019) suggest it is usually easier to extend the model ability to domains similar to the original learned data (*e.g.*, easier to generalize from trained *"dog faces"* to unseen *"human faces"*, rather than to unseen *"churches"*), we observe the completely opposite behavior. In this work, a bigger domain gap signifies a larger bandwidth for the unseen target domain, making it easier to sample from and less likely to be interfered and "captured" by the ID denoising trajectories, as presented in Tab. 2.

**Unseen Domain Image Synthesis.** Next, we show the quantitative results of our unseen domain image synthesis method in Tab. 3. We incorporate three groups of comparisons for evaluation. In the first group, we list the FID scores obtained via generative models trained as ID dataset, which can be considered as the upper bounds for this unseen image synthesis task. For the second group, we compare against other learning-based methods designed for image-to-image translation task. It is worth noting that those methods are not strictly doing data generation but rather in-domain image editing, since they do not sample from the unseen domain distribution. For the third group, we show the performance of our proposed method for various unseen domains using four different base models. We observe that our method achieves promising results comparable to the SOTA learning-based image translation approaches, demonstrating the effectiveness of our OOD latent geometric sampling.

**Visualization of Latent Encodings.** To provide a comprehensive understanding of the geometric locations for different OOD domains at the mixing step $t_m$, we include the t-SNE plot (Van der Maaten & Hinton, 2008) for the inverted latent encodings from different unseen image domains in Fig. 4. We plot the *"human"* and *"church"* as unseen OOD domains, and compare them with the ID *"dog"* domain, observing the consistent conclusion we draw from the bandwidth discussion, which argues that the domain with larger gaps compared to the originally trained one is easier to be distinguished and separated from the ID Gaussian distribution. In Fig. 4, we also illustrate two typical failure cases in our proposed *UnseenDiffusion* method. Specifically, the first failure case happens when the sampled OOD latent encoding passes the rejection selection but is located too close to the ID Gaussian distribution. Therefore, during the denoising process, the OOD unseen trajectory is interfered by the ID denoising trajectory which has a larger bandwidth, leading the final

denoised image to be more similar to the originally learned domain. The second failure case caused by the initial far-away location of sampled latent encoding can be usually mitigated by our geometric optimization. We provide additional ablation studies for geometric optimizations in Appendix F.

**Source of Generalization Ability.** Given our novel and somewhatcounterintuitive approach, one of the main questions is whether the model is in fact *generalizing* to the new domain. In other words: are the synthesized unseen images in fact *diverse and different from* the real images $x^{out}$ used for Gaussian estimation and geometric optimization? The above concern is seemingly related to the mode collapse problem where generative models like GANs tend to synthesize similar images. In fact, we explicitly clarify that the "mode collapse" does not exist in this work. The underlying cause of mode collapse is *model-dependent*, meaning that whichever latent encodings are drawn from the Gaussian prior, the trained model (*e.g.*, GANs) maps them to a similar data point in the raw image space. However, in this work, the arbitrary image reconstruction test ensures that the latent representation ability is **model-independent**. In extreme cases, there exists an infinite number of raw unseen images that can be always traced back

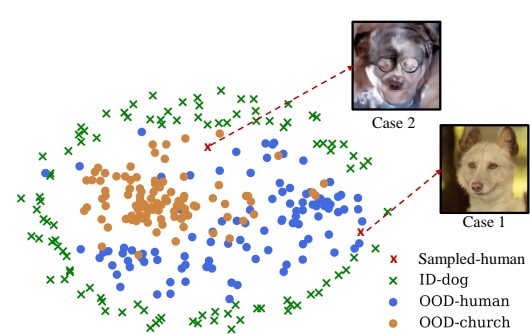

Figure 4: **T-sne visualization of ID and OOD latent encodings and failure cases demonstration.** The OOD latent encodings with larger domain gap (*e.g.*, *"church"*) are more separable compared to the unseen domain (*e.g.*, *"human faces"*) similar to the trained one (*e.g.*, *"dog faces"*), contradicting our traditional understanding from learning-based domain generalization methods. Typical failure cases happen when an OOD latent sample is too close to the ID Gaussian, leading to an ID image (case 1), or the latent sample is too isolated, leading to a low-fidelity image (case 2).

to high-dimensional latent spaces, the only challenge lies within the accurate localization of those potential latent encodings without knowing the corresponding denoised image in advance. Qualitative demonstrations showing the samples used for estimating and the synthesized ones are also included in Appendix F.

## 5 DISCUSSIONS AND CONCLUSION

**Why *UnseenDiffusion* works?** At the end of this paper, we seek to revisit and explore the underlying logic to explain the reasons for which our *UnseenDiffusion* works. From the recent *BoundaryDiffusion* work, the authors show that the semantic editing of images can be effectively achieved via *one-step* modification on the latent spaces after the semantic boundaries are localized, which reveals an interesting fact that the latent spaces of diffusion models preserve the semantic characteristics of raw data space from a geometric point of view. Specifically, if the images in the raw data space exhibit certain manifolds and distributions that can be semantically separated, it is very likely that those characteristics are preserved in the latent spaces of trained diffusion models, regardless of the training domains. On the other hand, our motivation of going towards latent spaces instead of staying in the raw space $\epsilon_0$ is the same as other generative models, which allows us to alleviate the difficulty of sampling for synthesizing new data in a relatively more regularized space with known distributions.

**Limitations and Broader Impact.** To summarize, we propose a novel perspective for rethinking the generalization ability for data synthesis using pre-trained diffusion models based on our inspiring theoretical and empirical findings. The limitation and bottleneck of this work come from the difficulties of localization of the target latent encodings. In other words, despite the theoretical Gaussian assumption and additional geometric support, it is still extremely hard to identify a valid OOD latent encoding without training. In fact, the ultimate goal of this work is *never performance-driven*. If one aims to generate high-fidelity nature images with low FID scores, we would not recommend using a DDPM trained on dog images as the tool. At the same time, we also acknowledge that because we adopt the unseen domain image synthesis as the downstream application, it poses the same risks of malicious use of synthetic data as other general generative works.

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

# A HIGH-LEVEL TAKE-AWAY AND APPENDIX OVERVIEW

## A.1 INSIGHTS ON SEVERAL OPEN QUESTIONS

From a very high-level perspective, our proposed *UnseenDiffusion* work features a heuristic-based method that attempts to provide some insights on several open questions in the generative domains with diffusion models:

- Where is the limit of representation ability for DDPMs?
- Where does the generalization ability of DDPMs come from?
- What role does the stochastic Gaussian noise play in diffusion models?

For the first question, our arbitrary image reconstruction tests demonstrate that each raw image can be deterministically traced back to the latent space and very well reconstructed. While we are not claiming that those latent representations are optimal (and apparently they are not), we can at least have the tool to trace the trajectories in both directions.

If the first question sets the foundation of the generalization ability of DDPMs to create *new and synthesized* data, the second question is less obvious and more difficult to provide empirical evidence. Recall the generative process in the unconditional DDPMs (after the training has been completed), there are essentially two steps, namely latent encoding sampling and denoising [7]. Given the extreme case, one can always find latent encodings that correspond to real images given a relatively fixed denoising trajectory, therefore the biggest challenge really comes to the first latent sampling step when synthesizing new data, *i.e.*, how to find a "good" latent encoding. To some degree, it seems that synthesizing new images, no matter ID or OOD image domains, is not a "creation" process, but rather a "discovery" process in the latent space.

**We note this is intrinsically different from the "mode collapse" issue in many GAN-based works.** The mode collapse describes the fact that *after a latent encoding is sampled*, the mapping trajectories collapse to a similar ending point in the image space. The intrinsic difficulty comes to "finding good trajectories", in contrast to the challenge of this work focuses on "finding good latent encodings".

Take a step forward to the third question on the role that the stochastic Gaussian plays within the diffusion framework. We notice some recent works touch on this interesting question from different angles. For instance, the *ColdDiffusion* (Bansal et al., 2022) empirically indicates that the stochastic Gaussian noises may not be necessary for diffusion models to generate new data. The *BoundaryDiffusion* (Zhu et al., 2023a) reveals the "distance effect", which exists only on the deterministic formulations that lead to distorted images. A potential unified answer to this open question based on our understanding is that stochastic noises may be a mitigation solution to relax the trade-off between sampling and denoising. The rationale behind this claim comes from the following aspects:

1) *Mitigation on the sampling*: according to our analysis and discussions on the bandwidth from Sec. 3.2, the model has to exhibit a certain level of tolerance on the stochasticity (*i.e.*, a non-zero bandwidth) to achieve unseen image synthesis, otherwise, it is extremely difficult (almost impossible from a probabilistic point of view) to sample a valid latent encoding.

2) *Mitigation on the denoising*: multiple existing literature (Ho et al., 2020; Song et al., 2021a; Zhu et al., 2023a; Karras et al., 2022) have proved that the stochasticity helps to improve the image quality. Specifically, Zhu et al. (2023a) provides a potential explanation from a geometrical point of view that the stochasticity helps to bring the OOD latent encodings back to the area with higher probabilistic concentration mass.

3) *A side note on the Gaussian assumption*: if all we need is stochasticity, how necessary is the Gaussian assumption? A potential answer to this question is: the Gaussian assumption may not be a hard request from the theoretical perspective, but is empirically very important. The key role of Gaussian assumption is to provide an easy interface to facilitate the sampling process, and has various important properties that can be used as mathematical tools to better interpret and understand the

---

[7]The denoising can also be considered as a mapping trajectory operation between the latent space and real data space.

model as well as the behaviors of latent encodings in the high-dimensional space. In theory, the latent space can also follow other distributions, but this would increase the difficulty to train the generative models and to do sampling during the inference stage.

## A.2 OVERVIEW

As authors of this work, we summarize the biggest contribution of this work is opening up a novel perspective to understand DDPMs, by providing abundant analysis and empirical results.

We structure the appendices as follows: In Appendix B, we present the detailed background of DDPMs. In Appendix C, we introduce more details about DDIMs, as well as the theoretical proof of the intermediate latent Gaussian assumption. In Appendix D, we provide the necessary background from the studies on high-dimensional spaces for understanding the geometric and spatial properties exhibited by the latent encodings, which sets the foundation for the geometric optimization in our work. Appendix E presents the spatial and geometric studies for diffusion models. More empirical results are presented in Appendix F.

## B  BACKGROUND ABOUT DDPMS

We briefly present the theoretical background of Denoising Diffusion Probabilistic Models (DDPMs) (Sohl-Dickstein et al., 2015; Ho et al., 2020) in the main paper, and describe more details in this section.

The objective of DDPMs is to similar to all the previous generative models, which is to approximate a data distribution $q(\mathbf{x}_0)$ with a learned model distribution $p_\theta(\mathbf{x}_0)$, from which we can easily sample from. Specifically, the original formulation considers the generative model in the following form:

$$p_\theta := \int p_\theta(\mathbf{x}_{0:T}) d\mathbf{x}_{1:T}, \tag{4}$$

where $\mathbf{x}_1, ..., \mathbf{x}_T$ are latent variables that represent the states of a Markov chain. At each step of the reverse process, the joint distribution is defined as a Markov chain with learned Gaussian transition:

$$p_\theta(\mathbf{x}_{t-1}|\mathbf{x}_t) := \mathcal{N}(\mathbf{x}_{t-1}; \mu_\theta(\mathbf{x}_t, t), \sum_\theta(\mathbf{x}_t, t)). \tag{5}$$

Particularly, the starting distribution is set to be a standard Gaussian in dimensionality $d$, with $p(\mathbf{x}_T) = \mathcal{N}(\mathbf{x}_T; \mathbf{0}, \mathbf{I})$.

Next, we have:

$$p_\theta(\mathbf{x}_{0:T}) := p_\theta(\mathbf{x}_T) \prod_{t=1}^{T} p_\theta^{(t)}(\mathbf{x}_{t-1}|\mathbf{x}_t). \tag{6}$$

For the pre-defined inference procedure $q(\mathbf{x}_{1:T}|q_0)$, DDPMs (Ho et al., 2020) propose to model this process using a Markov chain with Gaussian transitions parameterized by a decreasing sequence $\alpha_{1:T} \in (0, 1]^T$ as follows:

$$q(\mathbf{x}_{1:T}|\mathbf{x}_0) := \prod_{t=1}^{T} q(\mathbf{x}_t|\mathbf{x}_{t-1}), \tag{7}$$

with $q(\mathbf{x}_t|\mathbf{x}_{t-1}) := \mathcal{N}(\sqrt{\frac{\alpha_t}{\alpha_{t-1}}}\mathbf{x}_{t-1}, (1 - \frac{\alpha_t}{\alpha_{t-1}})\mathbf{I})$.

The training objective of the generative model $p_\theta(\mathbf{x}_0)$ is to optimize the variational lower bound on the negative log likelihood:

$$\begin{aligned}
\mathcal{L}_{vlb} &:= \mathbb{E}[-\log p_\theta(\mathbf{x}_0)] \\
&\leq \mathbb{E}[-\log \frac{p_\theta(\mathbf{x}_{0:T})}{q(\mathbf{x}_{1:T}|\mathbf{x}_0)}] \\
&= \mathbb{E}_q[-\log p(\mathbf{x}_T) - \sum_{t \geq 1} \log \frac{p_\theta(\mathbf{x}_{t-1}|\mathbf{x}_t)}{q(\mathbf{x}_t|\mathbf{x}_{t-1})}].
\end{aligned} \tag{8}$$

The terms "forward process" and "reverse process" are used to describe the transition from $x_0$ to $x_T$, and from $x_T$ to $x_0$, respectively.

# C  PROOF AND DERIVATIONS FOR LATENT DISTRIBUTIONS AFTER INVERSIONS

In this section, we provide detailed discussions on the inversion technique adapted from DDIMs (Song et al., 2021a), which is critical to our key observation on arbitrary image reconstruction tests. While the Gaussian assumption has been already proposed in DDIMs (but not utilized in their paper), we re-organize the logic and proof below for easy reference and for a better understanding of our work.

## C.1  DENOISING DIFFUSION IMPLICIT MODELS

While the original DDPMs involve a stochastic process for data generation via denoising (*i.e.*, the same latent encoding will output different denoised images every time after the same generative chain), there is a variant of diffusion model that allows us to perform the denoising process in a deterministic way, known as the Denoising Diffusion Implicit Models (DDIMs) (Song et al., 2021a). DDIMs were initially proposed for the purpose of speeding up the denoising process, however, later research works extend DDIMs from faster sampling application to other usages including the inversion technique to convert a raw image to its arbitrary latent space in a deterministic and tractable way. As briefly stated in our main paper, the core theoretical difference between DDIMs and DDPMs lies within the nature of forward process, which modifies a Markovian process to a non-Markovian one.

The key idea in the context of non-Markovian forward is to consider a family of $\mathcal{Q}$ of inference distributions, indexed by a real vector $\sigma \in \mathbb{R}_{\geq 0}^T$:

$$q_\sigma(\mathbf{x}_{1:T}|\mathbf{x}_0) := q_\sigma(\mathbf{x}_T|\mathbf{x}_0) \prod_{t=2}^{T} q_\sigma(\mathbf{x}_{t-1}|\mathbf{x}_t, \mathbf{x}_0), \tag{9}$$

where $q_\sigma(\mathbf{x}_T|\mathbf{x}_0) = \mathcal{N}(\sqrt{\alpha_T}\mathbf{x}_0, (1-\alpha_T)\mathbf{I})$ and for all $t > 1$,

$$q_\sigma(\mathbf{x}_{t-1}|\mathbf{x}_t, \mathbf{x}_0) = \mathcal{N}(\sqrt{\alpha_{t-1}}\mathbf{x}_0 + \sqrt{1-\alpha_{t-1}-\sigma_t^2} \cdot \frac{\mathbf{x}_t - \sqrt{\alpha_t}\mathbf{x}_0}{\sqrt{1-\alpha_t}}, \sigma_t^2\mathbf{I}). \tag{10}$$

The choice of mean function from Eqn. 10 ensures that $q_\sigma(\mathbf{x}_t|\mathbf{x}_0) = \mathcal{N}(\sqrt{\alpha_t}\mathbf{x}_0, (1-\alpha_t)\mathbf{I})$ for all $t$, so that it defines a joint inference distribution that matches the "marginals" as desired. The non-Markovian forward process can be derived from Bayes' rule:

$$q_\sigma(\mathbf{x}_t|\mathbf{x}_{t-1}, \mathbf{x}_0) = \frac{q_\sigma(\mathbf{x}_{t-1}|\mathbf{x}_t, \mathbf{x}_0)q_\sigma(\mathbf{x}_t|\mathbf{x}_0)}{q_\sigma(\mathbf{x}_{t-1}|\mathbf{x}_0)}. \tag{11}$$

In fact, in the original paper, the authors also explicitly stated that: " The forward process from Eqn. 11 is also Gaussian (although we do not use this fact for the remainder of this paper [8])". While this Gaussian property was not emphasized and leveraged in the DDIMs paper, we find it useful in our context to explore the representation and generalization ability of pre-trained DDPMs.

In particular, the hyper-parameters for Gaussian scheduling $\alpha$ and $\beta$ in the context of DDIMs are slightly different from the original formulation in DDPMs (Ho et al., 2020). Denote the original sequences from DDPMs as $\alpha'_t$, then the $\alpha_t$ in this work follows the definition of DDIMs to be $\alpha_t = \prod_{t=1}^{T} \alpha'_t$.

## C.2  PROOFS

The key theoretical support for the Gaussian assumption in the non-Markovian forward diffusion process is from Eqn. 2, where the marginal distribution of $\mathbf{x}_t$ given $\mathbf{x}_0$ satisfies a family of Gaussian.

---

[8]This paper refer to the DDIM paper (Song et al., 2021a).

**Lemma C.1.** *For $q_\sigma(\mathbf{x}_{1:T}|\mathbf{x}_0)$ defined in Eqn. 9 and $q_\sigma(\mathbf{x}_{t-1}|\mathbf{x}_t, \mathbf{x}_0)$ defined in Eqn. 10, we have:*

$$q_\sigma(\mathbf{x}_t|\mathbf{x}_0) = \mathcal{N}(\sqrt{\alpha_t}\mathbf{x}_0, (1 - \alpha_t)\mathbf{I}). \tag{12}$$

*Proof.* Assume for any $t \leq T$, $q_\sigma(\mathbf{x}_t|\mathbf{x}_0) = \mathcal{N}(\sqrt{\alpha_t}\mathbf{x}_0, (1 - \alpha_t)\mathbf{I})$ holds, if:

$$q_\sigma(\mathbf{x}_{t-1}|\mathbf{x}_0) = \mathcal{N}(\sqrt{\alpha_{t-1}}\mathbf{x}_0, (1 - \alpha_{t-1})\mathbf{I}), \tag{13}$$

then we can prove that the statement with an induction argument for $t$ from $T$ to 1, since the base case $(t = T)$ already holds.

First, we have that

$$q_\sigma(\mathbf{x}_{t-1}|\mathbf{x}_0) := \int_{\mathbf{x}_t} q_\sigma(\mathbf{x}_t|\mathbf{x}_0)q_\sigma(\mathbf{x}_{t-1}|\mathbf{x}_t, \mathbf{x}_0)d\mathbf{x}_t, \tag{14}$$

$$q_\sigma(\mathbf{x}_t|\mathbf{x}_0) = \mathcal{N}(\sqrt{\alpha_t}\mathbf{x}_0, (1 - \alpha_t)\mathbf{I}), \tag{15}$$

$$q_\sigma(\mathbf{x}_{t-1}|\mathbf{x}_t, \mathbf{x}_0) = \mathcal{N}(\sqrt{\alpha_{t-1}}\mathbf{x}_0 + \sqrt{1 - \alpha_{t-1} - \sigma_t^2} \cdot \frac{\mathbf{x}_t - \sqrt{\alpha_t}\mathbf{x}_0}{\sqrt{1 - \alpha_t}}, \sigma_t^2\mathbf{I}). \tag{16}$$

According to Bishop & Nasrabadi (2006) *2.3.3 Bayes' theorem for Gaussian variables*, we know that $q_\sigma(\mathbf{x}_{t-1}|\mathbf{x}_0)$ is also Gaussian, denoted as $\mathcal{N}(\mu_{t-1}, \Sigma_{t-1})$ where:

$$\mu_{t-1} = \sqrt{\alpha_{t-1}}\mathbf{x}_0 + \sqrt{1 - \alpha_{t-1} - \sigma_t^2} \cdot \frac{\sqrt{\alpha_t}\mathbf{x}_0 - \sqrt{\alpha_t}\mathbf{x}_0}{\sqrt{1 - \alpha_t}} = \sqrt{\alpha_{t-1}}\mathbf{x}_0, \tag{17}$$

$$\Sigma_{t-1} = \sigma_t^2\mathbf{I} + \frac{1 - \alpha_{t-1} - \sigma_t^2}{1 - \alpha_t}(1 - \alpha_t)\mathbf{I} = (1 - \alpha_{t-1})\mathbf{I}. \tag{18}$$

Therefore, $q_\sigma(\mathbf{x}_{t-1}|\mathbf{x}_0) = \mathcal{N}(\sqrt{\alpha_{t-1}}\mathbf{x}_0, (1 - \alpha_{t-1})\mathbf{I})$, which allows to apply the induction argument.

*Q.E.D*

If we take a closer look at the lemma and its proof, all the derivations are completed in the forward diffusion direction (*i.e.*, the inversion direction from data to latent space), and have not touched the trained model $p_\theta$, which sets the primary rationale to estimate the latent distributions as Gaussians, given raw data $\mathbf{x}_0$ as conditioning.

## D   HIGH-DIMENSIONAL GEOMETRIC OPTIMIZATIONS

The sole Gaussian assumption from *Lemma* is insufficient to practice unseen image synthesis in practice. On the one hand, there always exists a gap in the actual model training and its theoretical foundations, especially when we utilize the pre-trained DDPMs and have no control over the frozen model parameters. In other words, we can not guarantee that the base models we use perfectly model the Gaussians as expected trained using the variational loss. In fact, Zhu et al. (2023a) has recently demonstrated that the DDIM inversion does not exhibit symmetric diffusion trajectories as in actual generation, contradicting to previous understanding as in Kwon et al. (2023). On the other hand, we propose that a critical factor for a successful unseen image synthesis trail is for the denoising trajectory to stay uninterfered by the ID trajectories as illustrated in Fig. 3.

Given the above reasons, we seek to push forward the unseen image synthesis task by additional knowledge from a novel perspective from the geometric properties in high-dimensional space. Inspired by Zhu et al. (2023a), where the authors successfully achieve SOTA performance in image editing in a learning-free manner, **we notice that the diffusion models preserve certain geometric and spatial properties that characterize data semantics in training.** Therefore, we propose to explore the additional information of the unseen image domains by investigating their geometric properties in latent spaces.

| Domains | Pair-distance | Pair-Angle | Angle to Origin | Gaussian Radius | $dist(O_{in}, O_{out})$ |
|---|---|---|---|---|---|
| AFHQ-Dog(ID) | 608.3±3.2 | 60.0±0.4 | 90.1±0.3 | 430.5±2.2 | 0 |
| CelebA(OOD) | 577.1±5.0 | 60.0±0.5 | 89.7±0.4 | 410.4±4.2 | 71.1 |
| AFHQ-Cat(OOD) | 570.8±4.5 | 60.0±0.4 | 89.8±0.3 | 404.6±3.6 | 66.2 |
| AFHQ-Wild(OOD) | 574.1±4.0 | 60.0±0.4 | 89.8±0.3 | 406.6±3.6 | 63.9 |
| Bedroom(OOD) | 574.3±5.5 | 60.0±0.6 | 89.5±0.5 | 407.6±4.6 | 61.4 |
| Church(OOD) | 568.2±6.5 | 60.0±0.7 | 89.8±0.4 | 404.0±6.2 | 72.3 |
| CelebA(ID) | 609.2±3.4 | 60.0±0.5 | 90.0 ±0.3 | 432.4±2.7 | 0 |
| AFHQ-Dog(OOD) | 575.8±9.8 | 60.0±1.0 | 89.8±0.4 | 408.2±10.2 | 67.5 |
| Church(OOD) | 566.2±10.7 | 60.0±1.1 | 89.6±0.5 | 401.7±10.1 | 77.2 |

Table 4: **Geometric statistics for ID and different unseen OOD domains.** We summarize complete statistics we computed during our analytical experiments for different OOD domains, using the improved DDPM (Nichol & Dhariwal, 2021) trained on AFHQ-DOG-256 (Choi et al., 2020), and the DDPM (Ho et al., 2020) trained on CelebA-HQ (Karras et al., 2017) as the base models. Gaussian radius is another geometric metric adopted in Zhu et al. (2023a) for the empirical search of the mixing step. In addition, we also calculate the distance between the ID and OOD centers, denoted as $dist(O_{in}, O_{out})$, this can be used as further empirical justifications for the separability between different Gaussians.

### D.1 OOD GEOMETRIC PROPERTIES

As described in the main paper, while the theoretical derivations provide a Gaussian prior that facilitates the initial sampling, it is insufficient to generate unseen images in practice. To this end, we further incorporate geometric based optimizations to further ensure the successful rates of unseen image synthesis.

We consistently observe three geometric properties for the inverted OOD latent encodings. We provide a more detailed discussion on what each property implies in this sub-section.

Recall the three geometric properties as below:

**Observation 1:** *For any OOD sample pairs $\mathbf{x}_{inv,i}^{out}$ and $\mathbf{x}_{inv,j}^{out}$ from the sample set, the Euclidean distance between these two points is approximately a constant $d_o$.*

**Observation 2:** *For any three OOD samples $\mathbf{x}_{inv,i}^{out}$, $\mathbf{x}_{inv,j}^{out}$ and $\mathbf{x}_{inv,k}^{out}$ from the sample set, the angle formed between $\overrightarrow{\mathbf{x}_{inv,k}^{out}\mathbf{x}_{inv,i}^{out}}$ and $\overrightarrow{\mathbf{x}_{inv,k}^{out}\mathbf{x}_{inv,j}^{out}}$ is always around 60°.*

**Observation 3:** *For any OOD sample pairs $\mathbf{x}_{inv,i}^{out}$ and $\mathbf{x}_{inv,j}^{out}$ from the sample set, let O denote the origin in the high-dimensional space, the angle formed between $O\overrightarrow{\mathbf{x}_{inv,i}^{out}}$ and $O\overrightarrow{\mathbf{x}_{inv,j}^{out}}$ is always around 90°.*

For the first observation, when the sample pairs keep approximately the same distance, the direct implication is that those samples are likely to be drawn from some convex region in the high-dimensional space Wang (2012). One typical example is the spherical structure, where every data points exhibit an equal distance from the center.

The second geometric property suggests that the unknown samples could lie on a regular lattice near a low-dimensional manifold or sub-manifold, where the local geometry of the manifold is approximately Euclidean. However, a less evident implication is that for samples drawn from a high-dimensional Gaussian, this property also holds, as detailed in the next section D.2, and illustrated in Fig. 5(c).

The third geometry property implies that the sample points might be isotropic in nature, who are rotationally symmetric around any point in the space. Therefore, any two points drawn from the distribution are equally likely to lie along any direction in the space. This property is also observed for a high-dimensional Gaussian Blum et al. (2020), whose covariance matrix is proportional to the identity matrix.

We acknowledge that to deduce a distribution in high-dimensional space solely based on its geometric properties is very challenging, and there may exist other complex distributions that exhibit similar

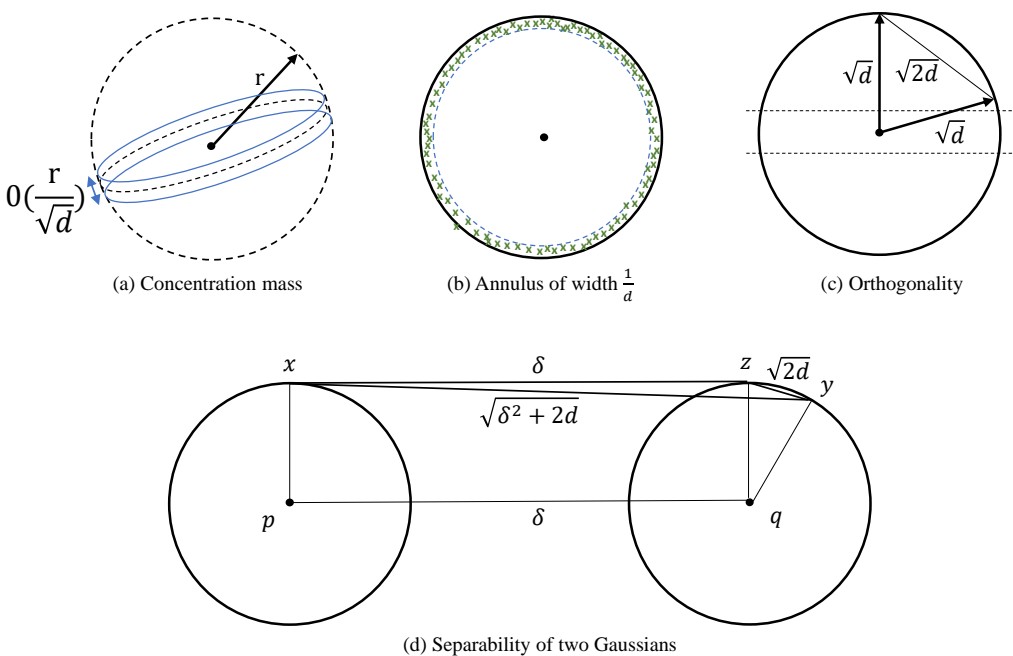

Figure 5: **Illustration of various geometric properties of high-dimensional Gaussians.** (a) and (b) show the probability concentration mass is mainly centered around a thin annulus around the equator, which are the properties mainly used in Zhu et al. (2023a). (c) illustrates the geometric observation on the orthogonality of sample pairs. (d) illustrates the idea of separating two Gaussian distributions in high-dimensional spaces.

properties we have observed. However, combined with our theoretical analysis and empirical observations, the OOD Gaussian assumption seems to hold well.

Explicitly, we find the above geometric properties do not hold for images $\mathbf{x}_0$ from the data space. For instance, the angle of samples to the origin is approximately $75°$ rather than $90°$.

## D.2 High-Dimensional Gaussian

Gaussian in high-dimensional space establishes various characteristic behaviors that are not obvious and evident in low-dimensionality. A better understanding of those unique geometric and probabilistic behaviors is critical to investigate the latent spaces of DDMs, since all the intermediate latent spaces along the denoising chain are Gaussian as demonstrated and proved in our previous sections.

We present below several properties of high-dimensional Gaussian from Blum et al. (2020), note those are known and established properties, we therefore omit the detailed proofs in this supplement, and ask readers to refer to the original book if interested.

***Property D.1.*** *The volume of a high-dimensional sphere is essentially all contained in a thin slice at the equator and is simultaneously contained in a narrow annulus at the surface, with essentially no interior volume. Similarly, the surface area is essentially all at the equator.*

This property above is illustrated in Fig. 5(a)(b), where the sampled ID encodings are presented in a narrow annulus.

***Lemma D.2.*** *For any $c > 0$, the fraction of the volume of the hemisphere above the plane $x_1 = \frac{c}{\sqrt{d-1}}$ is less than $\frac{2}{c}e^{-\frac{c^2}{2}}$.*

**Lemma D.3.** *For a d-dimensional spherical Gaussian of variance 1, all but $\frac{4}{c^2}e^{-c^2/4}$ fraction of its mass is within the annulus $\sqrt{d-1} - c \leq r \leq \sqrt{d-1} + c$ for any $c > 0$.*

The lemmas above imply that the volume range of the concentration mass above the equator is in the order of $O(\frac{r}{\sqrt{d}})$, also within an annulus of constant width and radius $\sqrt{d-1}$. In fact, the probability mass of the Gaussian as a function of $r$ is $g(r) = r^{d-1}e^{-r^2/2}$. Intuitively, this states the fact that the samples from a high-dimensional Gaussian distribution are mainly located within a manifold, which matches our second geometric observation.

**Lemma D.4.** *The maximum likelihood spherical Gaussian for a set of samples is the one over center equal to the sample mean and standard deviation equal to the standard deviation of the sample.*

The above lemma is used as the theoretical justification for the proposed empirical search method in Zhu et al. (2023a). We also adopt the search method using the Gaussian radius for identifying the operational latent space along the denoising chain to perform the OOD sampling.

**Property D.5.** *Two randomly chosen points in high dimension are almost surely nearly orthogonal.*

The above property corresponds to the *Observation 3*, where two inverted OOD samples consistently form a $90°$ angle at the origin.

### D.3 SEPARABILITY OF HIGH-DIMENSIONAL GAUSSIANS

Since our proposed latent geometric sampling method relies on the fact that the ID Gaussian distribution is separable from the OOD ones to ensure that the OOD denoising trajectories would not be captured and intervened by ID trajectories. We need to be able to theoretically support the underlying separability assumption. In statistics, separating Gaussians is also an established and formulated task. The statistical problem arises as to how much separation is needed between the means to tell which Gaussian generated which data point given a set of Gaussian distributions in high dimensionality.

In Blum et al. (2020), an example for separating two Gaussians in case of two spherical unit variance is given as illustration. We know that most of the probability mass of each Gaussian lies on an annulus of width $O(1)$ at radius $\sqrt{d-1}$ according to *Lemma 3*. Given two spherical unit variance Gaussians with centers $p$ and $q$ separated by a distance $\delta$, let $x$ and $y$ be two points randomly chosen from the first and second Gaussian, respectively. Their distance is close to $\sqrt{\delta^2 + 2d}$, since $x - p$, $p - q$, and $q - y$ are nearly perpendicular, as shown in Fig. 5(d).

The implicit requirement for separating the above two Gaussians is that the distance between two points picked from the same Gaussian is closer to each other than two points picked from different Gaussians. In other words, the upper limit of distance between a pair of points from the same Gaussian is an at most the lower limit of distance between points from different Gaussians, which holds when the following inequality is true:

$$\sqrt{2d} + O(1) \leq \sqrt{2d + \delta} - O(1). \tag{19}$$

To satisfy the above inequality in Eqn. 19, we need to ensure $\delta \in \Omega(d^{1/4})$.

Therefore, in statistics, we can safely separate mixtures of spherical Gaussians when their centers are separated by more than $d^{1/4}$. However, in practice, different Gaussians can even be separated when the centers are much closer.

In our analytical experiments, given DDMs trained on the image resolution of $3 \times 256 \times 256$, we have the total dimensionality to be $d = 196,608$, resulting in a safe separation distance to be $196,608^{1/4} \approx 21.1$. Based on our analytical experiments, we have a larger separation distance between ID and OOD Gaussians in Tab. 4, which provides theoretical and empirical justifications for the Gaussian separability test.

## E LATENT SPACES IN DIFFUSION MODELS

The understanding of latent spaces for diffusion models has evolved with the development of the field. While the latent spaces of DDMs are initially considered to lack semantic meanings Preechakul et al. (2022), several recent works Kwon et al. (2023); Zhu et al. (2023a) start to explicitly investigate the

---

**Algorithm 1** Estimation of pair-wise distance

---

**Input:** $N$ inverted OOD latent samples $x_{inv,t_m}^{out}$, number of sample pairs $n$ for estimation
**Output:** Pair-wise distance $d_o$
$dist \leftarrow 0$
**for** $i = 1, 2, ..., n$ **do**
  $(p, q) \leftarrow \text{RandomInt}(0, N - 1)$
  $dist+ = Euclidien\_distance(x_{inv,p}^{out}, x_{inv,q}^{out})$
**end for**
$d_o \leftarrow dist/n$

---

diffusion models from the perspective of high-dimensional latent spaces, and find that the pre-trained DDMs already have semantic spaces. In Zhu et al. (2023a), the authors explicitly point out that the deterministic inversion process is not symmetric to the actual denoising stochastic trajectory, and the semantic-meaningful behaviors are observed from different levels of latent spaces.

---

**Algorithm 2** Geometric Optimization

---

**Input:** A sampled OOD latent encoding $x_{sample}^{out}$, geometric pair-wise distance $d_{out}$, distance tolerance $d_{tol}$, angle tolerance $\varphi_{tol}$, $\mathcal{N}_{ref}$ OOD reference samples $x_{ref}^{out}$
**Output:** True or False
*// Step 1: Distance rejection based on Observation 1.*
**for** $i = 1, ..., N_{ref}$ **do**
  $d \leftarrow Euclidien\_distance(x_{ref,i}^{out}, x_{sample}^{out})$
  **if** $d < d_o - d_{tol}$ or $d > d_0 + tol$ **then**
    return $False$
  **end if**
**end for**
*// Step 2: Angle rejection based on Observation 2.*
**for** $i = 1, ..., N_{ref}$ **do**
  $(p, q) \leftarrow \text{RandomInt}(0, N_{ref} - 1)$
  $\varphi \leftarrow Angle(\overrightarrow{x_{sample}^{out} x_{ref,p}^{out}}, \overrightarrow{x_{sample}^{out} x_{ref,q}^{out}})$
  **if** $\varphi < 60 - \varphi_{tol}$ or $\varphi > 60 + \varphi_{tol}$ **then**
    return $False$
  **end if**
**end for**
*// Step 3: Angle rejection based on Observation 3.*
**for** $i = 1, ..., N_{ref}$ **do**
  $j \leftarrow \text{RandomInt}(0, N_{ref} - 1)$
  $\varphi \leftarrow Angle(\overrightarrow{Ox_{sample}^{out}}, \overrightarrow{Ox_{ref,j}^{out}})$
  **if** $\varphi < 90 - \varphi_{tol}$ or $\varphi > 90 + \varphi_{tol}$ **then**
    return $False$
  **end if**
**end for**
return $True$

---

### E.1 MIXING STEP IN DIFFUSION MODELS

Inspired from the Markov mixing time (Levin & Peres, 2017), Zhu et al. (2023a) introduces the concept of mixing step to characterize the convergence for diffusion models by considering the distance measure between the latent distribution and the stationary distribution in the Markov chain.

Specifically, they find that the mixing step is a generic feature formed in the training process of DDMs, which is related to the transition kernels, the stationary distribution, and the dimensionality of latent variables, described in the *Property* below (proof can be found in Zhu et al. (2023a)):

***Property E.1:*** *Under the total variation distance measure $|| \cdot ||_{TV}$, the mixing step $t_m$ for a DDM with data dimensionality $d$ is formed during training (i.e., irrelevant to the sampling methods).*

---

**Algorithm 3** *UnseenDiffusion* for Unseen Image Synthesis

---

**Input:** $N$ raw images from an unseen domain, a pre-trained DDM $p$, its mixing step $t_m$, total inversion steps $S_{inv}$, confidence level $c$, tolerance $tol$

**Output:** an image $x^{out}$ of the unseen domain.

*// Step 1: Obtain inverted OOD latent encodings at the mixing step $t_m$.*

Define $\{\tau_s\}_{s=1}^{S_{inv}}$ s.t. $\tau_1 = 0, \tau_{S_{inv}} = t_m$

**for** $i = 1, 2, ..., N$ **do**

    **for** $s = 1, 2, ..., S_{inv} - 1$ **do**

        $\epsilon \leftarrow p(\mathbf{x}_{i,\tau_s}, \tau_s)$

        $\mathbf{x}_{i,\tau_{s+1}} = \sqrt{\alpha_{\tau_s}}\mathbf{x}_{i,\tau_s} + \sqrt{1 - \alpha_{\tau_s}}\epsilon$

    **end for**

    Save the OOD latent $x_{i,\tau_{S_{inv}}}$ as $x_{inv,t_m}^{out}$

**end for**

$\mathcal{B}\eta \leftarrow$ Estimate the bandwidth from $x_{inv,t_m}^{out}$

$d_o \leftarrow$ Estimate the pair-wise distance from $x_{inv,t_m}^{out}$

*// Step 2: Fit a Gaussian directly using $x^{out}$ or $x_{inv,t_m}^{out}$ below.*

$\mu_{out}, \sigma_{out}^2 \leftarrow$ Mean, Cov($\{x_{1,inv}^{out}, x_{2,inv}^{out}, ..., x_{N,inv}^{out}\}$)

$\sigma_{md} \leftarrow std(dist(x_{i,inv}^{out}, x_{j,inv}^{out})_{\{i,j\}=\{1,...,N\}})$

*// Step 3: Sample and Geometric optimization.*

$x_{sample,t_m}^{out} \leftarrow \mathcal{N}(\mu_{out}, \sigma_{out}^2)$

**if** Rejected by $d_o$, $tol$, and angle criteria **then**

    Repeat sampling

**end if**

*// Step 4: Denoising via pre-trained DDM.*

$x^{out} \leftarrow p_{\eta = \mathcal{B}_\eta}(x_{sample,t_m}^{out})$

---

$t_m$ *is mainly related to the transition kernels, the stationary distribution (i.e., datasets), and the dimensionality $d$.*

We find the above idea very practical for exploring the operational latent space to do the OOD sampling, since one precondition in our task of unseen domain image synthesis is to ensure that the OOD distribution is distinguishable from the ID distribution to avoid the interference. We therefore adopt the empirical search method from Zhu et al. (2023a), and propose to select the latent space at the mixing step for performing the latent geometric sampling.

### E.2 GAUSSIAN ESTIMATIONS IN LATENT SPACES

The study of high-dimensional Gaussian distributions has been an important and long-lasting topic in mathematics and statistics.

In our main paper in Sec. 3.3, we are estimating the unseen Gaussians using the raw images $\mathbf{x}_0^{out}$ via Eqn. 2. However, this is not the only way to estimate the unseen Gaussians. As an alternative method, we can also estimate the mean and variance values in a statistical way, by calculating the mean and variance values directly from the inverted latent encodings $\mathbf{x}_{inv,t}^{out}$, instead of $\mathbf{x}_0^{out}$.

Generally speaking, given a set of sample points $x_1, x_2, ..., x_n$ in $d$ dimensionality [9], if we wish to fit those sample points using a spherical Gaussian $F$, and assuming the unknown Gaussian has the mean value $\mu$ and variance $\sigma^2$ in every direction (*i.e.*, isotropic). Then the probability of picking these very points from this Gaussian is given by:

$$c \exp(-\frac{(x_1 - \mu)^2 + (x_2 - \mu)^2 + ... + (x_n - \mu)^2}{2\sigma^2}), \tag{20}$$

where the normalizing constant $c$ is the reciprocal of $[\int e^{\frac{|x-\mu|^2}{2\sigma^2}} dx]^n$. The Maximum Likelihood Estimator (MLE) of this Gaussian $F$ is the one that maximizes the above probability in Eqn. 20.

---

[9]We adopt the notation $x$ without **bold** to represent the general case in statistics, which distinguishes from the notation $\mathbf{x}$ used for representing the actual images in this work.

Therefore, for a spherical Gaussian in high dimensionality, we can easily estimate the mean $\mu$ and variance $\sigma^2$ using the following two Lemmas. (Detailed proof can be found in the book (Blum et al., 2020))

**Lemma E.2.** *Let $\{x_1, x_2, ..., x_n\}$ be a set of $n$ points in d-space. Then $(x_1 - \mu)^2 + (x_2 - \mu)^2 + ... + (x_n - \mu)^2$ is minimized when $\mu$ is the centroid of the points $x_1, x_2, ..., x_n$, namely $\mu = \frac{1}{n}(x_1 + x_2 + ... + x_m)$.*

**Lemma E.3.** *The maximum likelihood spherical Gaussian for a set of samples is the one with center equal to the sample mean and standard deviation equal to the standard deviation of the sample.*

While both the method introduced in the main paper and this statistical method can be adopted for the Gaussian estimation, the former is slightly faster since there is no need to conduct the inversion for raw unseen images. However, as we claimed in the beginning in Appendix A, the Gaussian prior is just an empirical tool that facilitates the sampling. Actually, in our experiments, both ways result in a high rejection rate after the geometric optimization procedure. The above also aligns with our expectation, since the base model is pre-trained, there exists no hard restrictions or constraints to ensure the Gaussian condition. We also need to note that **finding a distribution for samples in high-dimensional spaces is a non-trivial problem that remains an unsolved challenge in statistics, and is out of the scope of this work**.

## F MORE EMPIRICAL DETAILS

### F.1 DETAILED ALGORITHMS

While we omit some implementation details in the main paper due to space limitations, we provide detailed versions of different steps taken in our proposed latent geometric sampling method. Algo. 1 is the estimation of pair-wise geometric distance, and Algo. 2 is the rejection sampling using the domain-specific geometric information as further optimizations. Algo. 3 summarizes the entire pipeline of our proposed *UnseenDiffusion* method.

### F.2 RESOURCES AND TIME COST

Our method does not involve any additional training or fine-tuning of the base diffusion models, therefore it is not heavily GPU dependent. However, as we have also mentioned in the Limitations and Bottleneck section in the main paper, the Gaussian fitting and sampling methods are relatively time-consuming. The time cost for the denoising process, however, depends on the skipping technique as in previous works Song et al. (2021a); Kim et al. (2022); Kwon et al. (2023); Zhu et al. (2023a), given our case of 60 denoising steps from $t_m = 500$, generating one image takes about $1.7 \sim 2.4$ seconds, on a single RTX-3090 GPU.

### F.3 DISCUSSION ON BANDWIDTH

As we mentioned in the main paper, we observe that the unseen domains with larger domain gaps have a larger bandwidth. In Fig. 6, we show additional qualitative results to demonstrate the above claim. In this example, we show more qualitative results for the bandwidth search in the reconstruction task, we select the maximum $\eta$ value that ensures the quality of reconstruction as the bandwidth for the specific unseen domain at a certain operational space of diffusion steps. Overall, the bandwidth is a hyper-parameter that relates to the base model and the unseen domains. We also draw an interesting conclusion here that the bandwidth also depends on the diffusion steps, showing that the bandwidth gets larger as the chain gets closer to the raw image domains. Our observations suggest that there exists a trade-off: while the bandwidth gets larger at the latent spaces closer to the raw image domains, sampling from OOD unseen distributions also gets more difficult.

### F.4 LEARNING-BASED BASELINES

In addition to the comparison with various SOTA diffusion-based image-to-image translation methods such as EGSDE Zhao et al. (2022), DiffusionClip Kim et al. (2022) and Asyrp Kwon et al. (2023),

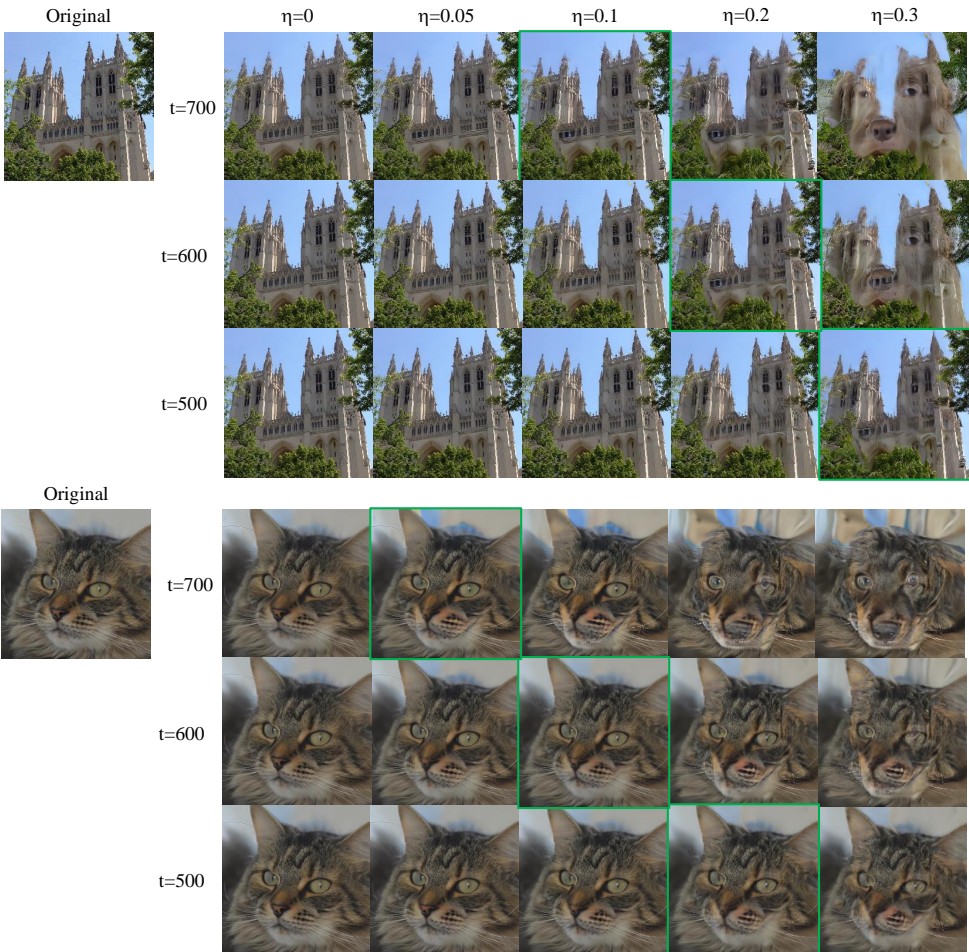

Figure 6: **Illustration of unseen trajectory bandwidth at different diffusion steps.** We show qualitative examples using the iDDPM Nichol & Dhariwal (2021) trained on AFHQ-Dog-256 as the base model, the examples of church and cat are both unseen domain images. The image in green boxes indicates the bandwidth we have empirically selected to preserve the reconstruction quality. Compared to the trained image domain (*i.e.*, *dogs*), *cats* have a smaller domain gap than *churches*. Different from the conventional understanding that a smaller domain gap is beneficial for better and easier generalization from a trained model, we observe a larger domain gap signifies a larger bandwidth, making it easier to perform the OOD sampling and synthesis.

we also conducted baseline experiments by fine-tuning the base DDM using the unseen images from different domains.

Specifically, we fined-tuned the iDDPM Nichol & Dhariwal (2021) trained on the AFHQ-Dog-256 dataset Choi et al. (2020) with 1K images from CelebA-HQ Karras et al. (2017) and LSUN-Bedroom-256 Yu et al. (2015), and show the qualitative experimental results in Fig. 7. We observe from Fig. 7 that the synthesized images after fine-tuning the model tend to preserve some features from the original domain, such as the furry hairs for human faces.

## F.5 MORE SYNTHESIS RESULTS

We present more qualitative results for the unseen image synthesis via our proposed latent geometric sampling in this section, in Fig. 8, Fig. 9 and Fig. 10.

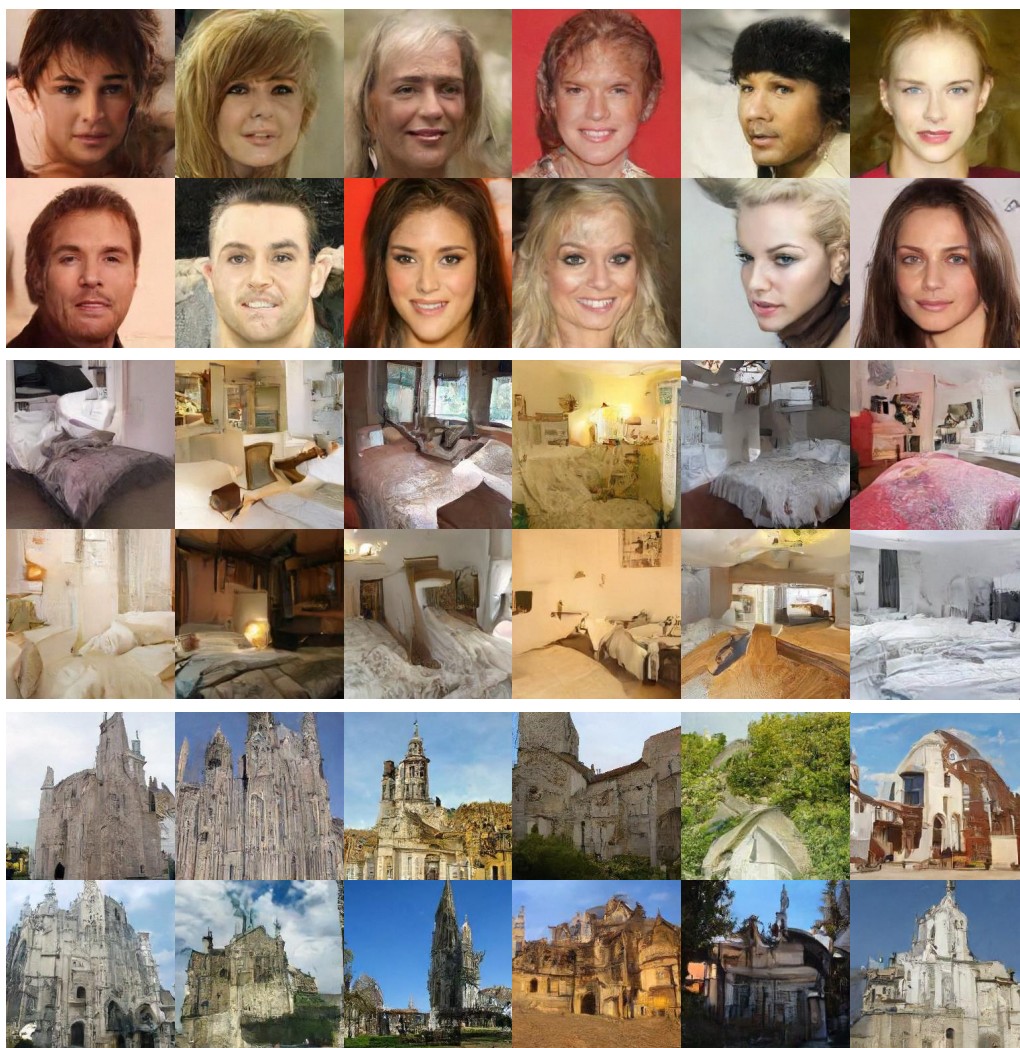

Figure 7: **Qualitative results from the fine-tuned diffusion models on different domains.** We show qualitative examples for the domain *human faces*, *bedrooms* and *churches* generated from fine-tuned diffusion models using the same amount of OOD images (1K).

The geometric optimization for the second round of rejection is important to improve the quality of sampled OOD latent encodings: it provides additional domain-specific information to further regularize the estimated Gaussian. We show the effect of geometric optimization on the synthesis performance in Tab. 5. The setting with $\mathcal{N}_{ref} = 1$ refers to only using the pair-wise distance $d_o$ as the rejection criterion (since the angle criterion requires at least two reference sample points).

Intuitively, more reference OOD samples for geometric optimization should be in general beneficial to guarantee and improve the quality of sampled latent encodings. In practice, we empirically observe $\mathcal{N}_{ref} = 3 \sim 4$ to be a reasonable number. With a larger number for $\mathcal{N}_{ref}$, we tend to increase the rejection rate, leading to a trade-off between synthesized quality and sampling difficulty.

## F.6 Empirical Results for Clarification on the Mode Collapse Issue

While we have clarified that the mode collapse is an intrinsically different problem that does not exist in this work, we also provide the raw OOD images used for unseen latent estimations and

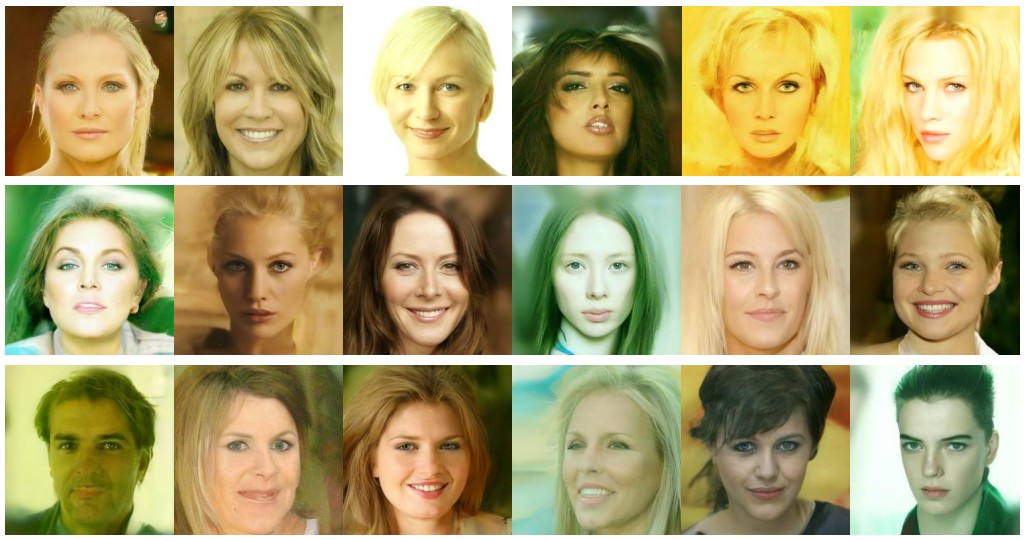

Figure 8: **More qualitative results from the unseen image synthesis task for the *human faces* image domain.** Using the iDDPM trained on AFHQ-Dog as the base model.

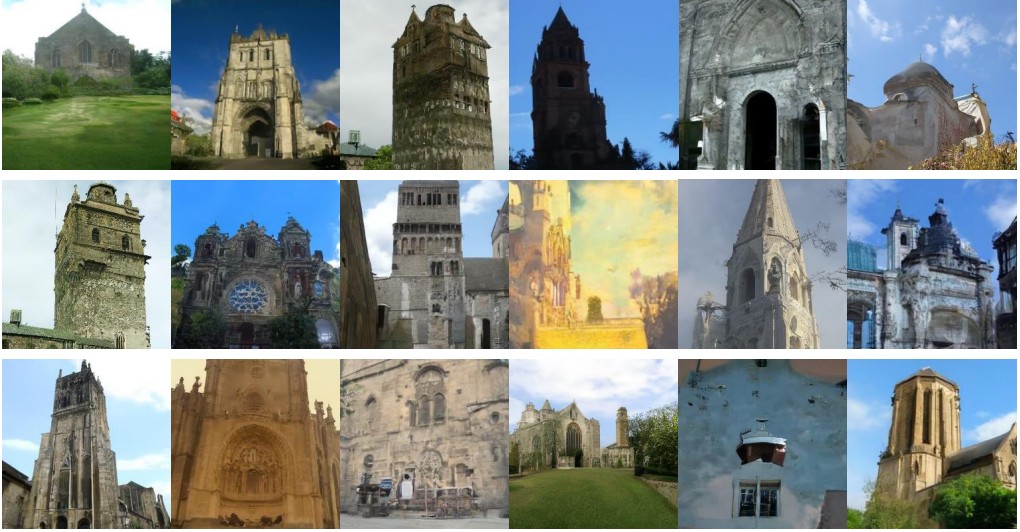

Figure 9: **More qualitative results from the unseen image synthesis task for the *churches* image domain.** Using the iDDPM trained on AFHQ-Dog as the base model.

geometric optimizations as empirical demonstrations in Fig. 11, Fig. 12, and Fig. 13. Compared to the synthesized images in their respective domains, we again demonstrate that the challenge in this work is different from "mode collapse" as in previous works.

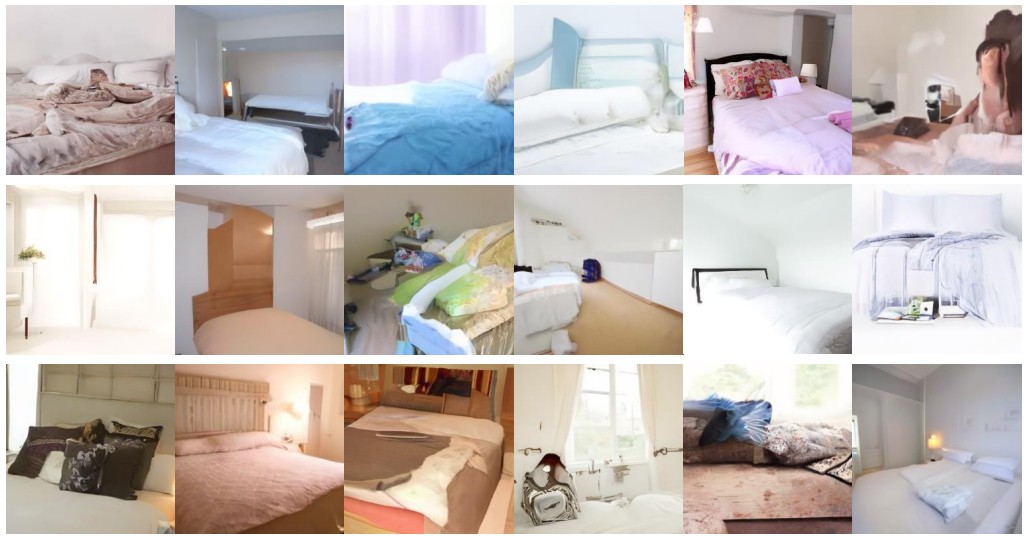

Figure 10: **More qualitative results from the unseen image synthesis task for the** *bedrooms* **image domain.** Using the iDDPM trained on AFHQ-Dog as the base model.

| OOD Domains | CelebA | Church | Bedroom |
|---|---|---|---|
| $\mathcal{N}_{ref} = 1$ | 47.7 | 46.4 | 45.1 |
| $\mathcal{N}_{ref} = 2$ | 44.5 | 45.2 | 44.8 |
| $\mathcal{N}_{ref} = 3$ | 43.8 | 44.6 | **43.3** |
| $\mathcal{N}_{ref} = 4$ | **43.5** | **43.9** | 43.4 |

Table 5: **Ablation on the effects of geometric optimization.** We report the FID scores for unseen domain image synthesis with different numbers of spatial reference OOD sample points. Note that we need a minimum of 2 OOD reference encodings to verify the pair-wise angle rejection criteria.

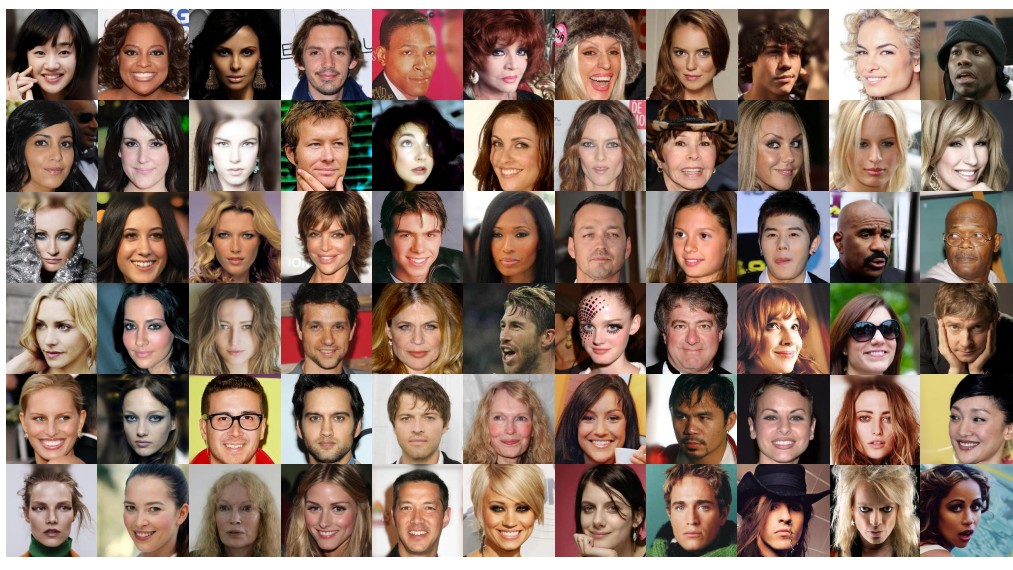

Figure 11: **Examples of raw human face images used for domain knowledge supplementary (*i.e.*, OOD Gaussian estimation and geometric optimizations) in our proposed *UnseenDiffusion*.**

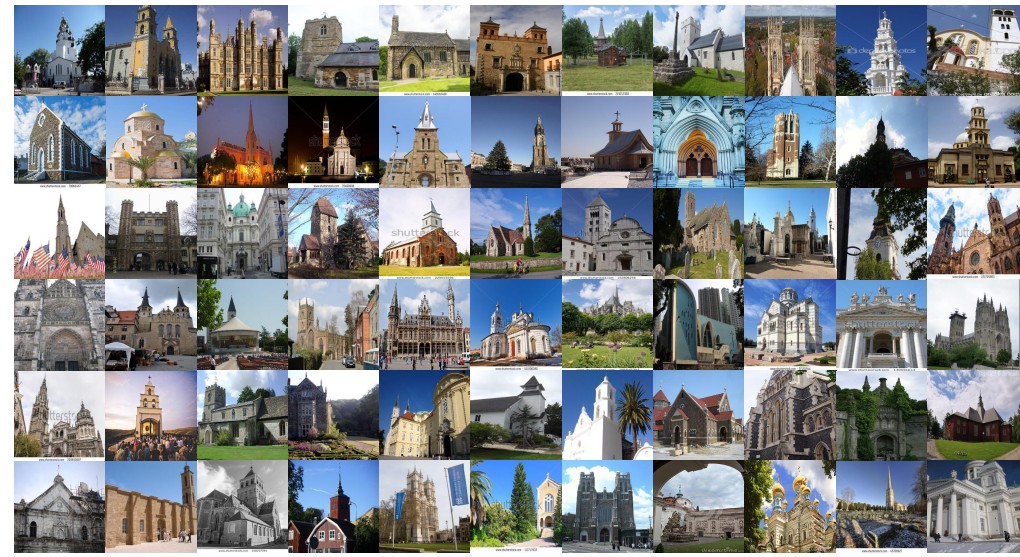

Figure 12: **Examples of raw church images used for domain knowledge supplementary (*i.e.*, OOD Gaussian estimation and geometric optimizations) in our proposed *UnseenDiffusion*.**

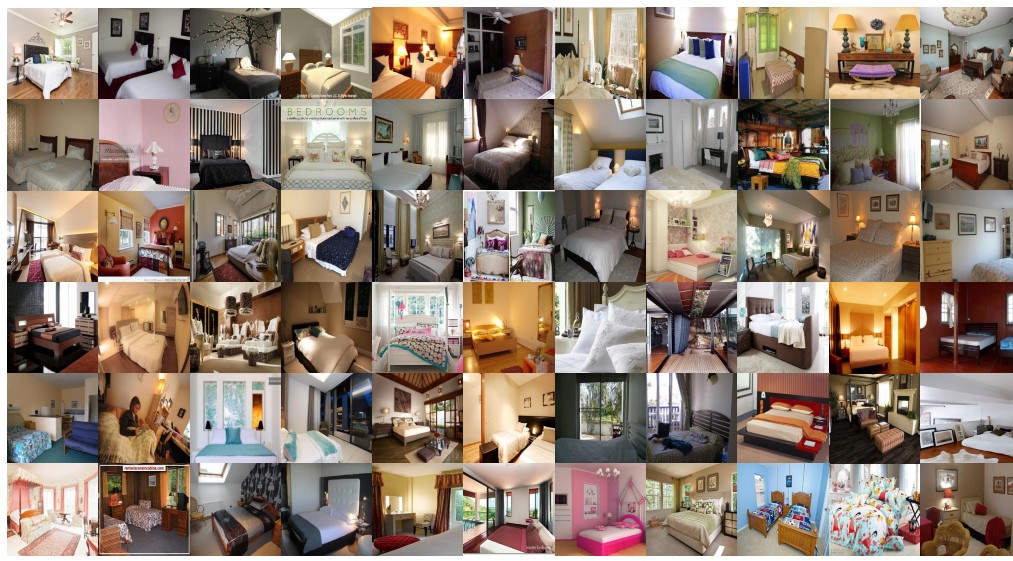

Figure 13: **Examples of raw bedroom images used for domain knowledge supplementary (*i.e.*, OOD Gaussian estimation and geometric optimizations) in our proposed *UnseenDiffusion*.**

