# OpenReview forum: "Unseen Image Synthesis with Diffusion Models"
_ICLR.cc/2024/Conference — ICLR 2024 Conference Withdrawn Submission_

### Official Review · Reviewer_2gqd · 2023-10-28

**Soundness:** 3 good
**Presentation:** 3 good
**Contribution:** 3 good
**Rating:** 6
**Confidence:** 4

**Summary:**

This manuscript elucidates a set of intriguing viewpoints, reconceptualizing the generative mechanism of diffusion models through a geometrical and spatial lens within the latent space. Although a significant proportion of these insights draws inspiration from [1], this study proffers a series of innovative concepts. For instance, it advocates for an intuitive exploration of the trajectory's bandwidth, leveraging both untrained and post-training models, subsequently deducing the model's capacity for OOD generation.


[1] Zhu Y, Wu Y, Deng Z, et al. Boundary guided mixing trajectory for semantic control with diffusion models[J]. arXiv preprint arXiv:2302.08357, 2023.

**Strengths:**

Through a meticulous examination and scrutiny of the latent space inherent to diffusion models, the author has derived several intriguing findings. Stemming from these observations, a bold hypothesis was posited: OOD inherently conforms to a Gaussian distribution. The author leverages these three distinct observations as criteria to discern the quintessential OOD distribution. Upon juxtaposition with certain image editing methodologies, it becomes evident that the sampling approach, anchored in geometric perspectives, is adept at generating authentic OOD samples. Collectively, this manuscript, building upon the foundations laid by [1], delves deeper into the associated phenomena and furnishes salient insights.


[1] Zhu Y, Wu Y, Deng Z, et al. Boundary guided mixing trajectory for semantic control with diffusion models[J]. arXiv preprint arXiv:2302.08357, 2023.

**Weaknesses:**

This article leans more towards a technical discussion. Its ability to generate OOD samples lacks competitiveness compared to models trained in relevant domains. At the same time, many fundamental viewpoints in the article are based on other works, making the overall article lack originality.

**Questions:**

My principal reservations regarding this study pertain to the bandwidth aspect. While it serves as an empirical parameter that seems to counter intuitive understanding, the manuscript commendably delineates numerous properties associated with it. Yet, a palpable void exists in terms of a rigorous quantitative analysis of this parameter. Moreover, the author's quest for latent encoding necessitates the inclusion of specific OOD samples, positioning this approach closer, in conceptual terms, to methodologies like few-shot learning or domain adaptation.

Additionally, regarding the issue of asymmetry between the forward and backward processes mentioned in the related literature, could it be caused by the schedule rather than the model itself?

---

### Official Review · Reviewer_RR6j · 2023-10-30

**Soundness:** 3 good
**Presentation:** 3 good
**Contribution:** 2 fair
**Rating:** 3
**Confidence:** 4

**Summary:**

The paper proposes UnseenDiffusion, a method to study the generalization ability of diffusion models like DDPM. The method is based on the observation that any image, regardless of its nature, can be made generated by a diffusion model by finding its latent representation via DDIM. Departing from this latent representation, allowing some stochasticity to the reverse DDPM process, the approach makes it possible to generate new unseen images that remind the given sample. This way, provided N samples of a new OOD domain, the approach can fit a new Gaussian distribution of latent variables that lead to the generation of plausible images of unseen OOD domain, without re-training the DDPM model.

**Strengths:**

- The paper proposes an original way to study the generalization of diffusion models.
- The results show an interesting ability of diffusion models to be able to generate new images of an unseen domain, even if this domain is distant from the dataset on which DDPM was originally trained.
- The paper is written well and the ideas are easy to follow.

**Weaknesses:**

Although the paper provides some original ideas and analysis, I was not convinced by the main claims.

- $\underline{\text{Generalization}}$. To claim generalization, it is required to demonstrate that the "new" generated images of OOD domain are significantly different from the $N$ real images used to estimate the gaussian distribution. At the moment, the paper does not convincingly demonstrate that this is indeed the case. While there is a discussion in the context of "mode collapse", it only suggests that generated images are different from each other (and not from $N$ provided images). In general, my intuition is that the "new" generated images look good if their latent representation is close to one of $N$ computed real OOD representations, but can be significantly deteriorated in quality between them. Another aspect that strengthens this concern is the FID of 43+ compared to the upper bounds of 6-8, which demonstrates that the general quality of produced "new" images is not great (which is not a sign good generalization).
   - To extend the analysis the generalization, I think it would be interesting to see a batch of uncurated OOD-generated images put aside their nearest neighbours based on some measure like LPIPS. Another interesting vizualization would be a trajectory of generated images from latent point of one real image to another (e.g., similar to latent space interpolations in GANs).

- $\underline{\text{Claims in bold}}$. I also have questions regarding the insights offered in the paper.
   - $\textit{"DDPM ... has sufficient representation ability to reconstruct arbitrary unseen images"}$. This is hardly an observation, since this ability is constructed directly through the deterministic inversion via DDIM. It is not yet a sign of generalization, since the model can still just reproduce the provided $N$ images or their slight variations.
   - $\textit{"... an interesting fact that contradicts conventional wisdom"}$. This claim is supported only by different values $B$ that were picked manually by hand, based on my understanding. Apart from the fact that manual handpicking is subject to biases, it is not clear how statistically significant is the difference between 0.2 and 0.3 in Table 2. Lastly, the $B$ values themselves are not yet indicators of generalization, as they only show how easy it is to reconstruct a provided image, not how to generate new OOD examples that differ.

Lastly, I would suggest the authors to move the Algorithm 3 to the main paper, since it is in essence the main technical implementation step contains the proposed method.

**Questions:**

Please reply to the concerns raised in Weaknesses.

---

### Official Review · Reviewer_NgdU · 2023-10-31

**Soundness:** 3 good
**Presentation:** 3 good
**Contribution:** 2 fair
**Rating:** 6
**Confidence:** 4

**Summary:**

The paper introduces a technique for generating Out-of-Distribution (OOD) images utilizing a pre-trained diffusion model. For example, the model, initially trained on the AFHQ dataset and having never encountered human faces, can be employed to generate such images. The outlined approach begins by acquiring the inverted latent vectors from N OOD images through DDIM inversion, subsequently constructing empirical heuristics from these vectors. Leveraging these heuristics allows for the intelligent sampling necessary to generate OOD images.

**Strengths:**

> The paper exhibits an innovative approach by targeting the generation of images with reduced training datasets or entirely Out-of-Distribution (OOD) images, which is an engaging deviation from the norm.

> The paper is well-structured and articulately written, with the findings being supported by solid experimental evidence.

> The unexpected results highlighted in the paper demonstrate the potential efficacy of the proposed method in the domain of OOD image generation.

> The method's ability to leverage pre-existing diffusion models to generate OOD images showcases a resourceful utilization of pre-trained models, which could be seen as a step towards more efficient generative modeling.

>The integration of Geometric Optimization (Algorithm 2) indicates a methodical approach to addressing the challenges associated with generating OOD images. This also results in interesting empirical observations.

**Weaknesses:**

> The proposed method, while innovative, employs existing techniques to identify latent vectors for OOD image generation, which may not be seen as groundbreaking. Along with this, the process outlined in Algorithm 2 (Geometric Optimization) for rejecting and re-sampling latent from empirical mean and covariance resembles the approach of fitting a basic machine learning model to produce these latent vectors, which might not be perceived as a novel solution.

> The paper lacks a detailed discussion on the practical implications and applications of the proposed method, particularly in real-world scenarios like text-to-image generation, which could potentially limit its broader relevance and utility. The paper could benefit from a more thorough exploration of the practical use cases or potential domains where the proposed method could be effectively employed.

> The process of generating OOD images, as described, follows a logical sequence of steps from inverting images to identifying the conducive region for OOD image generation, which may not evoke surprise among readers familiar with such techniques.

**Questions:**

> How can one use this for conditional diffusion models, such as text-to-image diffusion models?

> Can one fit a simple Machine-learning model to sample effectively from the latent space that will generate OOD images?

> Is it possible to do this for other generative models like say VQ-VAE, GANs by inverting the images?

---

### Official Review · Reviewer_zhbU · 2023-11-02

**Soundness:** 2 fair
**Presentation:** 2 fair
**Contribution:** 3 good
**Rating:** 5
**Confidence:** 4

**Summary:**

This paper proposes a way to generate images from unseen domain with a pretrained diffusion model by some heurustic rules.

**Strengths:**

1. The topic studied in this paper is interesting. It is suprising to see that the diffusion model trained with dog images can be used to generate images on human face and bedrooms.

2. The observations proposed in this paper is also interesting, showing that there exists some rules for the diffusion model.

**Weaknesses:**

1. The organization. Algorithm 1,2, 3 or something similar need to be put in the main paper.  Section 3 jumps to the experiments suddenly.

2. The experiments and observations. Although the observations are quite interesting, it needs more experiments to verify this point. The paper focuses on the tuple <dog, face, bedroom, church>.  Given this paper is based on empirical observation, more evidences need to be shown that the observations hold.

3. Some other metrics need to be included, such as diversity and dissimilarity to the original target images. Becasuse a trivial solution can be using the inverted noise to generate the images, which is just the reconstruction of the target domain. So, the FID can be very close to 0 even the proposed algorithm fails totally.

4. The rejection rate of the algorithm need to be discussed. Authors have three rules to reject the sampled latents from a high-dimensional Gaussian prior.  The computation time and rejection rates need to be discussed.

**Questions:**

as above